# Sharpness-Aware Pretraining Mitigates Catastrophic Forgetting

**Ishaan Watts**,* **Catherine Li**,* **Sachin Goyal, Jacob Mitchell Springer & Aditi Raghunathan**
Carnegie Mellon University
Pittsburgh, PA 15213, USA
`{iwatts,catheri4,sachingo,jspringe,aditirag}@andrew.cmu.edu`

## Abstract

Standard optimizer choices for pre-training are designed to minimize pre-training loss. Yet pre-trained models are routinely subjected to further transformations—such as fine-tuning to acquire new capabilities or quantization for efficiency. In this work, we evaluate optimizer choices across model scales, token budgets, and datasets, and find that strategies that explicitly (Sharpness-Aware Minimization) or implicitly (large learning rates and Warmup–Stable–Decay schedules) reduce sharpness yield better downstream performance, even when they achieve comparable or worse pre-training loss. Combining these strategies yields a new pre-training recipe that substantially outperforms standard baselines with minimal compute overhead, delivering a better learning–forgetting frontier during fine-tuning and higher accuracy after quantization.

## 1 Introduction

Large language models are built via pre-training followed by post-training adaptation such as fine-tuning or quantization. A core assumption is that lower pre-training loss implies better post-training performance. Recent evidence of catastrophic overtraining challenges this view: beyond a point, more pre-training can *worsen* downstream results despite improving base-model loss (Springer et al., 2025). This motivates optimization choices that target downstream robustness rather than pre-training loss alone.

We study how pre-training optimization choices shape downstream behavior through the lens of loss-landscape sharpness. Across 80+ pre-training runs and 3500+ fine-tuning runs, we find that sharpness-aware choices consistently improve adaptation. In particular, Sharpness-Aware Minimization (SAM) Foret et al. (2021) yields a better learning–forgetting frontier than AdamW Loshchilov & Hutter (2019) and reduces degradation under quantization and random perturbations. We also find that higher learning rates and Warmup-Stable-Decay (WSD) schedules Hu et al. (2024) improve downstream performance relative to cosine schedules Loshchilov & Hutter (2017), even when pre-training loss is slightly worse.

The two interventions we identify—SAM and learning rate scheduling—are in fact synergistic. We synthesize them to propose **Sharpness-Aware Warmup-stable Decay (SAWD)**, which applies SAM only during the decay phase of WSD, achieving most of SAM's benefits at minimal compute overhead. Overall, our results show that optimizing for sharpness yields pre-trained checkpoints that are more robust to post-training shifts, and SAWD provides a simple, practical recipe for better downstream performance.

## 2 Preliminaries

Optimizer choices are typically evaluated by pre-training loss, but our focus is on how pre-training affects downstream behavior after model modifications.

---

*Equal contribution.

## 2.1 Downstream properties of pre-trained models

Let $\theta_{\mathrm{PT}}$ denote a pre-trained model. We study two downstream properties.

**Learning–forgetting tradeoff.** Fine-tuning improves task performance but can degrade pre-trained knowledge (Goodfellow et al., 2013; Kirkpatrick et al., 2017; Springer et al., 2025). We measure learning via the fine-tuning loss $\mathcal{L}_{\mathrm{FT}}(\theta_{\mathrm{FT}})$ and forgetting via the pre-training loss evaluated at fine-tuned weights $\mathcal{L}_{\mathrm{PT}}(\theta_{\mathrm{FT}})$. Because results depend on fine-tuning hyperparameters, we summarize each base model by the Pareto frontier:

$$\left\{ \left( \mathcal{L}_{\mathrm{FT}}(\theta_{\mathrm{FT}}), \mathcal{L}_{\mathrm{PT}}(\theta_{\mathrm{FT}}) \right) \mid \theta_{\mathrm{FT}} \in \Theta_{\mathrm{FT}}(\theta_{\mathrm{PT}}) \right\},$$

where $\Theta_{\mathrm{FT}}(\theta_{\mathrm{PT}})$ represents the set of all models obtained by fine-tuning $\theta_{\mathrm{PT}}$ under varying fine-tuning configurations.

**Post-train quantization.** Quantization maps weights to lower precision, yielding $\widetilde{\theta_{\mathrm{PT}}}$. We evaluate degradation via $\mathcal{L}_{\mathrm{PT}}(\widetilde{\theta_{\mathrm{PT}}})$.

## 2.2 Sharpness as an approximation for forgetting

Let $\theta_{\mathrm{PT}} + \Delta$ denote the result of a downstream modification. A second-order expansion of the pre-training loss gives

$$\mathcal{L}_{\mathrm{PT}}(\theta_{\mathrm{PT}} + \Delta) \approx \mathcal{L}_{\mathrm{PT}}(\theta_{\mathrm{PT}}) + \nabla \mathcal{L}_{\mathrm{PT}}(\theta_{\mathrm{PT}})^\top \Delta + \tfrac{1}{2} \Delta^\top H \Delta,$$

where $H := \nabla^2 \mathcal{L}_{\mathrm{PT}}(\theta_{\mathrm{PT}})$ is the Hessian of the pre-training loss. After long pre-training, the gradient is small and forgetting is dominated by curvature:

$$\mathcal{L}_{\mathrm{PT}}(\theta_{\mathrm{FT}}) - \mathcal{L}_{\mathrm{PT}}(\theta_{\mathrm{PT}}) \approx \tfrac{1}{2} \Delta^\top H \Delta.$$

This approximation links higher curvature (sharpness) to greater forgetting, motivating optimizers that control sharpness.

## 2.3 Optimization recipes to induce flatness

We consider two mechanisms for inducing flatness: an *explicit* approach via Sharpness-Aware Minimization (SAM) and an *implicit* approach via learning rate dynamics.

**Sharpness-Aware Minimization (SAM).** SAM Foret et al. (2021) searches for minima that remain low-loss under parameter perturbations: Given the pre-training objective $\mathcal{L}_{\mathrm{PT}}(\theta)$ and a radius $\rho > 0$, SAM solves the robust optimization problem:

$$\min_{\theta} \max_{\|\epsilon\|_2 \leq \rho} \mathcal{L}_{\mathrm{PT}}(\theta + \epsilon). \tag{1}$$

In practice, SAM performs a short ascent step followed by a descent step at the perturbed point.

**Learning rates and the Edge of Stability.** The Edge of Stability Cohen et al. (2021) suggests learning rate $\eta$ implicitly caps sharpness, with dynamics hovering at $\lambda_{\max}(H) \approx 2/\eta$. Implying that learning-rate magnitude and annealing duration can strongly affect forgetting.

## 3 Experiments

In this section, we evaluate catastrophic forgetting for two sharpness-aware approaches: explicit minimization with SAM and implicit control through learning-rate schedules.

### 3.1 Experimental Setup

**Pretraining.** We pre-train OLMo models (Groeneveld et al., 2024) at 20M, 60M, and 150M parameters on 4B–192B tokens from DCLM-Baseline (Li et al., 2024), using AdamW or SAM with cosine or WSD schedules. We tune learning rates for AdamW+cosine to minimize pre-training loss, reuse them for all settings, and set $\rho = 0.05$ for SAM (Appendix C.1).

**Fine-tuning.** We fine-tune on five datasets (StarCoder Li et al. (2023), GSM8K Cobbe et al. (2021), StackMathQA Zhang (2024), Tülu Lambert et al. (2025), MusicPile Yuan et al. (2024)) with AdamW+cosine, sweeping learning rates $1 \times 10^{-6}$—$1 \times 10^{-2}$ at batch size 64 for one epoch or 10M tokens (Appendix C.2). Results for only two datasets—StarCoder (code generation) and GSM8K (math reasoning)—are shown in the main paper. For each base model, we evaluate the learning–forgetting frontier by measuring fine-tuning loss and pre-training loss on validation data.

## 3.2 THE LEARNING-FORGETTING FRONTIER OF SAM VS ADAMW

Our first set of experiments compare pre-training with SAM and AdamW using the standard *cosine* learning rate for both optimizers.

**Token-matched setting.** We compare SAM and AdamW across model sizes, token budgets, and datasets. SAM consistently yields pre-trained checkpoints that forget less when fine-tuned to the same performance as AdamW counterparts (Figure 1a) For example, in StarCoder, SAM **reduces forgetting by** $80\%$ at matched fine-tuning levels (limiting pre-train loss growth to $+0.1$ vs. $+0.5$ for AdamW). SAM checkpoints also achieve strictly lower fine-tuning loss, showing they can learn more while achieving a better tradeoff. Furthermore, SAM's advantage persists and often widens with more tokens and larger models (Appendix D.2.2 and D.2.4).

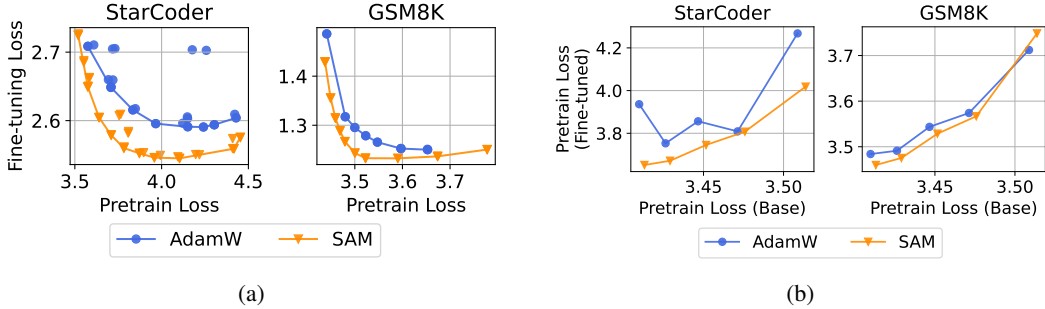

(a)

(b)

Figure 1: **SAM yields better learning–forgetting frontiers than AdamW (a) and mitigates catastrophic-overtraining (b).** OLMo-60M models pre-trained on 192B token and then fine-tuned.

**Pre-train loss matched setting.** Traditionally, AdamW is reported to optimize more aggressively than SAM, raising the question of whether SAM's benefits stem simply from 'lagging behind' on the optimization trajectory—effectively acting as a form of early stopping to avoid brittle, over-trained regimes. To rule this out, we compare optimizers at matched pre-train loss. We plot the retained pre-training loss after fine-tuning (subject to a fixed downstream threshold) against the initial loss (Figure 1b) [1]. SAM strictly dominates even in this loss-matched setting: for any given initial loss, SAM checkpoints retain more knowledge than AdamW counterparts. We also recover catastrophic overtraining for AdamW Springer et al. (2025): as the pre-trained model improves, retained pretrain loss eventually rises, indicating "better" base models paradoxically do worse after fine-tuning. In contrast, SAM exhibits a monotonic trend, suggesting the onset of catastrophic overtraining is pushed to significantly lower pretrain loss.

## 3.3 EFFECT OF LEARNING RATE SCHEDULES ON THE LEARNING–FORGETTING FRONTIER

As motivated in Section 2, learning-rate dynamics shape sharpness. We find that larger peak learning rates improve downstream tradeoffs even when pre-training loss is slightly worse (Appendix D.3). Beyond peak learning rate, the temporal structure of decay impacts model properties. By maintaining high learning rate for most of training, WSD implicitly regularizes the solution, preventing sharpening associated with decaying rates (Figure 2a). We further ablate the annealing fraction and find longer annealing windows lead to worse frontiers (Appendix D.3).

---

[1] Formally, we plot the minimum pre-training loss $\mathcal{L}_{\text{PT}}$ achievable subject to $\mathcal{L}_{\text{FT}}(\theta_{\text{FT}}) < \tau$. More details in Appendix C.3.

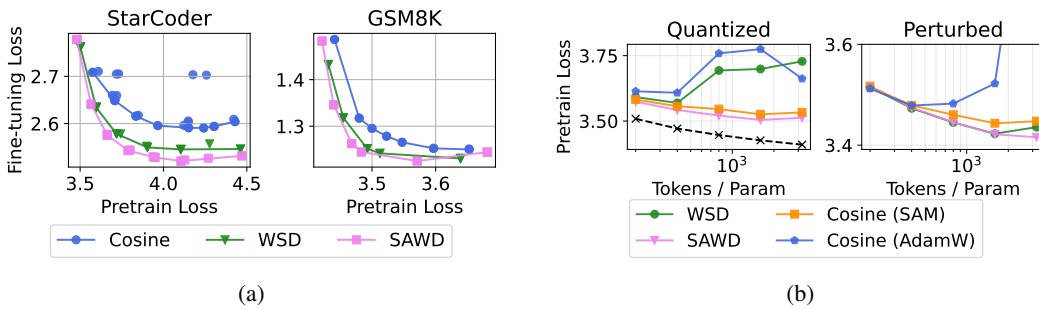

Figure 2: **WSD mitigates forgetting over Cosine schedule (a). SAM, WSD improve stability to weight perturbations like 4bit-quantization and Gaussian noise (b).** OLMo-60M models pre-trained on: 192B tokens and fine-tuned (a), 4B-192B and quantized/perturbed (b).

### 3.4 BEYOND FINE-TUNING: ROBUSTNESS TO WEIGHT PERTURBATIONS

Fine-tuning is the most common setting in which a pre-trained checkpoint is *modified*, but not the only one. Before deployment, practitioners routinely perturb model weights via quantization, pruning, or compression. We test robustness under two common post-training perturbations—*4-bit quantization* and *isotropic Gaussian weight noise*—and measure the increase in pre-training validation loss. Sharpness-aware pre-training reduces degradation in both settings, with SAM and WSD consistently outperforming AdamW+cosine (Figure 2b). The gains are strongest in the high-token regime where quantization can otherwise induce catastrophic overtraining. Full details and plots are provided in Appendix D.5 and D.6.

## 4 HOW TO OVERTRAIN YOUR LANGUAGE MODEL

Our results suggest that sharpness-aware interventions are especially valuable in the overtraining regime where small models train on large token budgets ($N/D \gg 20$) for inference efficiency.[2] SAM offers robust gains over AdamW, but its two-step update doubles per-iteration cost. Meanwhile, our schedule experiments show that WSD already provides better downstream performance than cosine by maintaining a high learning rate during the stable phase, with only the brief annealing period—where the learning rate drops—responsible for increased brittleness. This suggests we can capture SAM's benefits precisely where they are needed most: during annealing. We therefore propose **Sharpness-Aware Warmup-stable Decay (SAWD)**, which applies SAM updates only during the decay phase of WSD, keeping compute overhead negligible.

Across learning–forgetting, quantization, and perturbation robustness, SAWD consistently outperforms AdamW and AdamW+WSD (Figure 2), yielding strong gains with negligible extra cost.

## 5 CONCLUSION

We show that optimizer choices during pre-training—both algorithms and learning-rate schedules—affect downstream performance beyond pre-training loss. Landscape sharpness emerges as a key factor linking pre-training dynamics to downstream adaptability: both explicit sharpness minimization (SAM) and implicit regularization via learning-rate dynamics (WSD, higher learning rates) produce checkpoints robust to fine-tuning and quantization. We combine these insights in SAWD, a simple and efficient pre-training recipe that yields consistent downstream improvements.

Our study is empirical and focuses on fine-tuning and simple post-training quantization. Future work could explore more advanced quantization methods and reinforcement learning for alignment. More broadly, we hope this work encourages a rethinking of pre-training optimization. While optimization choices are heavily tuned and standardized, our results show that evaluating downstream performance rather than pre-training loss can outperform AdamW, the long-standing default—highlighting significant potential for optimizers that target what ultimately matters: downstream performance.

---

[2]The recent OLMo-2 7B was pre-trained on nearly 4T tokens (OLMo et al., 2024).

## SCIENCE OF DL IMPROVEMENT CHALLENGE SUBMISSION

### WHAT MODEL ARE YOU TARGETING?

*Provide a summary of the problem the deep net model is designed to solve. Good summaries should outline the state of the literature, provide an overview that domain experts would consider reasonable, and cite relevant sources.*

Large language models (LLMs) are commonly trained under the assumption that lower pre-training loss implies better downstream performance. Recent work challenges this assumption, showing that beyond a threshold, continued pre-training can *degrade* post-training performance despite improving pre-training loss—a phenomenon known as *catastrophic overtraining* (Springer et al., 2025). Consequently, standard optimization practices based on AdamW (Loshchilov & Hutter, 2019) and cosine learning-rate schedules (Loshchilov & Hutter, 2017) fail to account for downstream robustness. This work addresses the problem of identifying pre-training optimization strategies that yield models robust to the parameter shifts induced by post-training modifications. Motivated by links between learning dynamics (Cohen et al., 2021), loss landscape sharpness (Foret et al., 2021), and generalization, we favor flatter minima during pre-training to reduce degradation under fine-tuning and quantization. This problem is most acute in the *overtraining regime*, where small models are trained on very large token budgets ($N/D \gg 20$) for inference efficiency, a setting increasingly common in practice (e.g., OLMo-2-7B trained on nearly 4T tokens (OLMo et al., 2024)). We therefore reframe pre-training from minimizing loss alone to optimizing the *entire model lifecycle*.

### HOW DO YOUR RESULTS CONTRIBUTE—OR COULD POTENTIALLY CONTRIBUTE—TO UNDERSTANDING THESE MODELS?

*What aspects of the models become better understood thanks to your work?*

Our work establishes how pre-training optimization choices impact downstream performance through the lens of loss-landscape sharpness. We show that downstream performance improves under optimization strategies that reduce sharpness—either explicitly via Sharpness-Aware Minimization (SAM) (Foret et al., 2021) or implicitly through learning-rate dynamics such as larger learning rates and Warmup–Stable–Decay schedules (Hu et al., 2024)—even when pre-training loss is comparable or slightly worse. Explicit sharpness-aware updates produce pre-trained checkpoints that are more tolerant to parameter perturbations, yielding a superior learning–forgetting frontier during fine-tuning and increased robustness to post-training quantization. Implicitly, learning-rate schedules act as a form of sharpness regularization: extended low-learning-rate annealing tends to drive models toward sharp minima that are fragile under adaptation. Together, these findings explain why lower pre-training loss does not necessarily translate to better downstream performance and establish sharpness as a meaningful and predictive property of pre-trained models.

### HOW DO YOU EXPECT YOUR SUBMISSION TO INFLUENCE FUTURE WORK?

*Propose ways in which your insights, findings, or methodologies could shape subsequent research directions, model design choices, or scientific applications.*

We anticipate that our insights will reshape standard pre-training practices while inspiring further research on downstream robustness. Practically, we introduce **Sharpness-Aware Warmup–Stable–Decay (SAWD)**, a simple and compute-efficient pre-training recipe that combines WSD with a brief SAM phase during learning-rate decay. SAWD consistently outperforms standard AdamW with cosine scheduling in downstream fine-tuning and quantization, and can be readily adopted in existing large-scale pre-training pipelines. Methodologically, our evaluation framework treats downstream robustness—fine-tuning, quantization, and parameter perturbations—as a first-class criterion for pre-training recipes, encouraging optimization over the full model lifecycle rather than pre-training loss alone. Scientifically, our findings motivate future work on scalable sharpness estimation, deeper theoretical understanding of how learning-rate schedules influence landscape geometry, and exploring whether sharpness-based principles extend to other modalities or to alignment methods such as reinforcement learning. More broadly, we hope our work motivates the field to prioritize downstream adaptability and knowledge retention as critical pre-training objectives.

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

## A    RELATED WORK

**Catastrophic forgetting and continual learning.**    The challenge of learning from sequential data without overwriting previously acquired knowledge is classically studied as *catastrophic forgetting* (Goodfellow et al., 2013). Continual learning proposes mitigation strategies including (i) *regularization* to protect important parameters (Kirkpatrick et al., 2017; Zenke et al., 2017; Aljundi et al., 2018); (ii) *replay* of past examples (Shin et al., 2017); (iii) *gradient constraints* (Lopez-Paz & Ranzato, 2017; Chaudhry et al., 2018); or (iv) *architectural* expansion (Rusu et al., 2016). These methods target explicit task sequences; we instead study forgetting *under post-training modifications* as a property of the pretrained checkpoint induced by long-horizon optimization.

**Loss of plasticity and catastrophic overtraining.**    A closely related phenomenon is *loss of plasticity*: with continued optimization, networks can become harder to adapt even when training loss continues to improve. This effect has been documented in deep reinforcement learning as *primacy bias* (Nikishin et al., 2022) and in supervised settings where warm-starting can impede later training dynamics (Ash & Adams, 2020). In large language models, catastrophic overtraining formalizes a similar outcome: additional pre-training can make models less amenable to downstream fine-tuning by increasing update sensitivity (Springer et al., 2025). We build on this line by connecting plasticity loss to sharpness growth under extended pre-training, and by showing that sharpness-aware and schedule-aware pre-training can preserve post-training stability.

**Optimization geometry: sharpness and sharpness-aware training.**    A long line of work connects generalization to loss-landscape geometry, including sharpness (Keskar et al., 2016), though interpretation requires care due to parameterization effects (Dinh et al., 2017). Modern training often operates near a critical *edge of stability* where learning rate and Hessian eigenvalues jointly govern dynamics (Cohen et al., 2021). SAM directly targets neighborhood robustness by optimizing for parameters whose perturbations do not significantly increase loss (Foret et al., 2021). We study how extended pre-training changes sharpness and how that controls forgetting under post-training interventions.

**Learning-rate schedules and long-horizon training.**    Learning-rate schedules are a primary control for stability and geometry over long training horizons. Cosine schedules with warm restarts are widely used and serve as a standard baseline (Loshchilov & Hutter, 2017). Warmup–Stable–Decay

(WSD) schedules have been motivated as practical long-horizon recipes and analyzed through a loss-landscape lens that links schedule structure to stability across training phases (Wen et al., 2024). Our results complement these insights by showing that schedule choices interact with sharpness growth under extended training, and that combining schedule design with sharpness-aware pre-training improves post-training stability.

# B  DEFINITIONS

## B.1  OPTIMIZERS

Optimizers are algorithms used to update a model's parameters during training by minimizing a chosen loss function. Given gradients of the loss with respect to model parameters, optimizers determine the direction and magnitude of parameter updates. Their design plays a critical role in training stability, convergence speed, and generalization performance.

### B.1.1  ADAMW

AdamW is an adaptive gradient-based optimizer that combines the benefits of momentum and per-parameter learning rate adaptation, while decoupling weight decay from the gradient update. This decoupling corrects a flaw in standard Adam and leads to improved generalization.

Let $g_t = \nabla_\theta \mathcal{L}(\theta_t)$ denote the gradient at step $t$. AdamW maintains exponential moving averages of the first and second moments:

$$m_t = \beta_1 m_{t-1} + (1 - \beta_1)g_t, \quad v_t = \beta_2 v_{t-1} + (1 - \beta_2)g_t^2.$$

After bias correction,

$$\hat{m}_t = \frac{m_t}{1 - \beta_1^t}, \quad \hat{v}_t = \frac{v_t}{1 - \beta_2^t}.$$

The parameter update is given by

$$\theta_{t+1} = \theta_t - \alpha \frac{\hat{m}_t}{\sqrt{\hat{v}_t} + \epsilon} - \alpha \lambda \theta_t,$$

where $\alpha$ is the learning rate and $\lambda$ is the weight decay coefficient.

### B.1.2  SAM

Sharpness-Aware Minimization (SAM) is an optimization framework designed to improve generalization by explicitly favoring flatter minima. Unlike AdamW, which updates parameters based on gradients at the current point, SAM seeks parameters whose neighborhood exhibits uniformly low loss. This results in solutions that are less sensitive to small perturbations and empirically generalize better.

At each step, SAM first computes a perturbation in the direction of the gradient:

$$\epsilon_t = \rho \frac{\nabla_\theta \mathcal{L}(\theta_t)}{\|\nabla_\theta \mathcal{L}(\theta_t)\|_2},$$

where $\rho$ controls the size of the neighborhood. The parameters are temporarily perturbed to $\tilde{\theta}_t = \theta_t + \epsilon_t$, and the final update is computed using the gradient at this perturbed point:

$$\theta_{t+1} = \theta_t - \alpha \nabla_\theta \mathcal{L}(\tilde{\theta}_t).$$

By optimizing for robustness within a local neighborhood, SAM encourages convergence to flatter minima, which has been shown to yield improved generalization compared to standard optimizers such as AdamW.

## B.2  LEARNING RATE SCHEDULERS

Learning rate schedules control how the optimizer's step size evolves during training and play a crucial role in convergence stability and generalization. Large learning rates enable rapid progress early in training, while smaller learning rates later help stabilize convergence. Consequently, modern training setups rarely use fixed learning rates, instead opting for dynamic learning rates defined by a schedule.

### B.2.1 Cosine

The cosine schedule gradually anneals the learning rate following a cosine curve, allowing for large updates early in training and smaller, more stable updates near convergence. It is commonly combined with a warmup phase, during which the learning rate increases linearly from zero to the peak learning rate $\alpha_{\text{max}}$ over $T_{\text{warmup}}$ steps. After warmup, the learning rate follows the cosine decay:

$$\alpha_t = \alpha_{\text{min}} + \frac{1}{2}(\alpha_{\text{max}} - \alpha_{\text{min}}) \left( 1 + \cos \left( \frac{\pi(t - T_{\text{warmup}})}{T - T_{\text{warmup}}} \right) \right),$$

where $\alpha_{\text{min}}$ is the minimum learning rate, often set to zero, $T$ is the total number of training steps, and $t > T_{\text{warmup}}$.

### B.2.2 Warmup–Stable–Decay

This warmup–stable–decay (WSD) schedule consists of three phases: an initial warmup phase where the learning rate increases linearly from zero to a peak value, a stable phase where the learning rate is held constant, and a final decay/anneal phase where the learning rate is gradually reduced. Warmup mitigates optimization instability caused by large gradients at initialization, the stable phase enables effective learning at a fixed scale, and the decay phase promotes convergence.

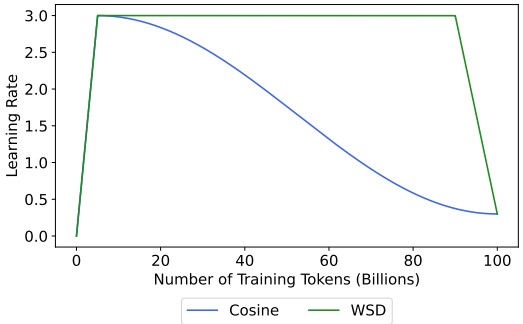

Figure 3: **Learning Rate Schedule for Cosine and WSD**

## C Experimental Details

### C.1 Pretraining

We tune the learning rate for each model checkpoint individually to minimize the pre-training validation loss. For SAM, we select $\rho = 0.05$ (defined in Section 2.3), which we determined by tuning preliminary small-scale model checkpoints.

### C.1.1 Model Configuration

We pre-train models from scratch at three parameter scales: 20M, 60M, and 150M using the OLMo architecture Groeneveld et al. (2024) and pre-training recipe. The model configurations can be seen in Table 1. Each model is trained with varying token budgets (Table 2), corresponding to token to parameter ratios of 100 to 3200, on the DCLM web data Li et al. (2024). We evaluate two optimizers, AdamW Loshchilov & Hutter (2019), and Sharpness-Aware Minimization Foret et al. (2021) in combination with two learning rate schedules: cosine Loshchilov & Hutter (2017) and warmup-stable-decay (WSD) Hu et al. (2024).

### C.1.2 LR Tuning

We tune learning rate of the pretrained models with AdamW and Cosine schedule and use the best value found for SAM (which uses AdamW as the base optimizer) and WSD schedule. For each

Table 1: Pre-training hyperparameters used in our controlled experiments.

| Hyperparameters | OLMo-20M | OLMo-60M | OLMo-150M |
|---|---|---|---|
| Layers | 8 | 8 | 12 |
| Heads | 8 | 8 | 12 |
| Vocab Size | 100278 | 100278 | 100278 |
| Embedding Size | 100352 | 100352 | 100352 |
| Hidden dimensions | 256 | 528 | 768 |
| Max context length | 1024 | 1024 | 1024 |
| Activation type | SwiGLU | SwiGLU | SwiGLU |
| Attention dropout | 0.0 | 0.0 | 0.0 |
| Residual dropout | 0.0 | 0.0 | 0.0 |
| Embedding dropout | 0.0 | 0.0 | 0.0 |
| Beta1 | 0.9 | 0.9 | 0.9 |
| Beta2 | 0.95 | 0.95 | 0.95 |
| Warmup steps | 1000 | 2000 | 3000 |
| Weight decay | 0.1 | 0.1 | 0.1 |
| Batch size | 256 | 256 | 256 |

Table 2: Pre-training token budgets used in our controlled experiments.

| Model | Token Budgets (B) |
|---|---|
| OLMo-20M | $4, 8, 16, 32, 64$ |
| OLMo-60M | $12, 24, 48, 96, 192$ |
| OLMo-150M | $15, 30, 60, 120$ |

model size, we start with the smallest token budget and sweep over the learning rates $\in [1e-4, 3e-4, 6e-4, 1e-3, 3e-3, 1e-2]$ to find the one which has the lowest pre-training loss on a held out validation set. For the next token budget, we only look at smaller LR's to check it's better based on observation from past work which shows optimal learning rate decreases with increasing token budgets Bjorck et al. (2025). Our final learning rates used for each combination of model size and tokens per parameter can be seen in Table 3 and the corresponding schematic of tuning can be seen in Figure 4.

Table 3: Final pre-training learning rates for different model sizes for varying tokens to parameter ratios. We use the same pre-training learning rate with SAM (which uses AdamW as the base optimizer) and with WSD schedule.

| Tokens / Param | 20M | 60M | 150M |
|---|---|---|---|
| 100 | – | – | 0.001 |
| 200 | 0.003 | 0.001 | 0.001 |
| 400 | 0.003 | 0.001 | 0.0006 |
| 800 | 0.003 | 0.0006 | 0.0003 |
| 1600 | 0.001 | 0.0006 | – |
| 3200 | 0.001 | 0.0003 | – |

## C.2 FINE-TUNING

We finetune the different pre-trained checkpoints on five publicly available datasets: StarCoder Li et al. (2023) (code generation), GSM8K Cobbe et al. (2021) and StackMathQA Zhang (2024) (mathematical reasoning), Tülu Lambert et al. (2025) (instruction following) and MusicPile Yuan et al. (2024) (domain-specific). We use the AdamW optimizer with a cosine learning rate schedule. To estimate the learning-forgetting tradeoff set, we sweep over learning rates ranging from $1e-6$ to $1e-2$. We tune batch size and weight decay to find 64 and 0 as optimal values respectively. We

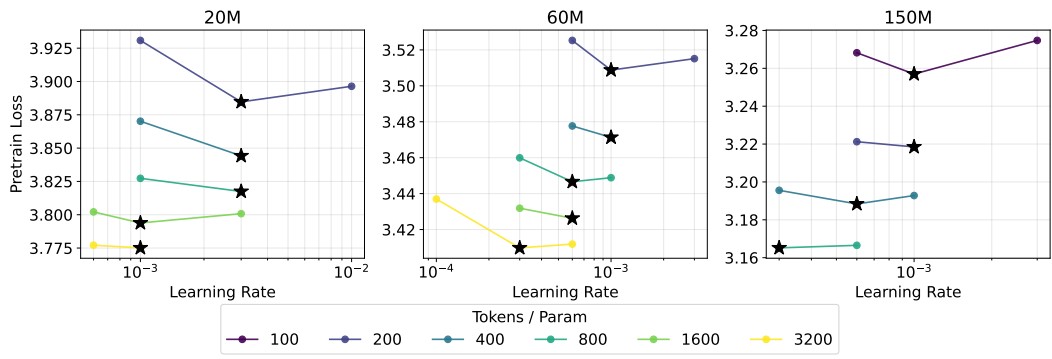

Figure 4: Pretrain learning rate tuning

finetune on each dataset for one epoch or a maximum of 10M tokens. This results in 1 epoch for StackMathQA (1.2M) and 10M for all four other datasets.

Table 4: Fine-tuning hyperparameters used in our controlled experiments.

| Hyperparameters | Values |
|---|---|
| Learning rate | $1e{-}6, 2e{-}6, 4e{-}6, 8e{-}6, 1e{-}5, 2e{-}5, 3e{-}5, 4e{-}5, 5e{-}5, 6e{-}5, 7e{-}5,$ $8e{-}5, 9e{-}5, 1e{-}4, \quad 1.1e{-}4, 1.2e{-}4, 1.25e{-}4, 1.4e{-}4, 1.5e{-}4, 1.6e{-}4,$ $1.8e{-}4, 2e{-}4, 2.4e{-}4, 2.5e{-}4, 3e{-}4, 3.5e{-}4, 4e{-}4, 5e{-}4, 6e{-}4, 7e{-}4,$ $8e{-}4, 9e{-}4, 1e{-}3, 1.25e{-}3, 1.5e{-}3, 2e{-}3, 2.5e{-}3, 3e{-}3, 4e{-}3, 5e{-}3,$ $6e{-}3, 8e{-}3, 1e{-}2$ |
| Batch size | 32, 64*, 128 |
| Learning rate scheduler | Cosine |
| Optimizer | AdamW |
| Weight decay | 0.0*, 0.1 |
| Warmup steps | 10% of training |
| Tokens | Min(tokens in dataset, 10M) |

## C.3 EVALUATION

For a given pre-trained checkpoint $\theta_{\text{PT}}$, we define the learning–forgetting tradeoff set as

$$\mathcal{T}(\theta_{\text{PT}}) = \left\{ \left( \mathcal{L}_{\text{FT}}(\theta_{\text{FT}}), \mathcal{L}_{\text{PT}}(\theta_{\text{FT}}) \right) \mid \theta_{\text{FT}} \in \Theta_{\text{FT}}(\theta_{\text{PT}}) \right\}.$$

To enable loss-matched comparison across pre-trained checkpoints, we define a common fine-tuning loss threshold as follows. For each checkpoint $\theta_{\text{PT}}^{(i)}$, we first compute the minimum fine-tuning loss achieved within its tradeoff set:

$$\mathcal{L}_{\text{min}}^{(i)} = \min_{(\mathcal{L}_{\text{FT}}, \mathcal{L}_{\text{PT}}) \in \mathcal{T}(\theta_{\text{PT}}^{(i)})} \mathcal{L}_{\text{FT}}.$$

We then define the global fine-tuning threshold $\tau$ as the maximum over these per-checkpoint minima:

$$\tau = \max_{i} \mathcal{L}_{\text{min}}^{(i)}.$$

For each pre-trained checkpoint, we report the retained pre-training loss $\mathcal{L}_{\text{PT}}$ corresponding to the model on its tradeoff frontier whose fine-tuning loss satisfies $\mathcal{L}_{\text{FT}} \leq \tau$.

# D    ADDITIONAL RESULTS

## D.1    PRETRAIN LOSS

On a token-matched setting, we see that SAM indeed lags behind AdamW on small models (20M, 60M). However, on larger model sizes ($\approx$150M), we see that SAM matches or even slightly outperforms AdamW in just the raw pre-train loss for the same number of tokens (Figure 5). In this work, our focus is on downstream properties of the pre-trained model, but it's interesting future work to evaluate rigorously if SAM can also achieve better pre-train loss to start with.

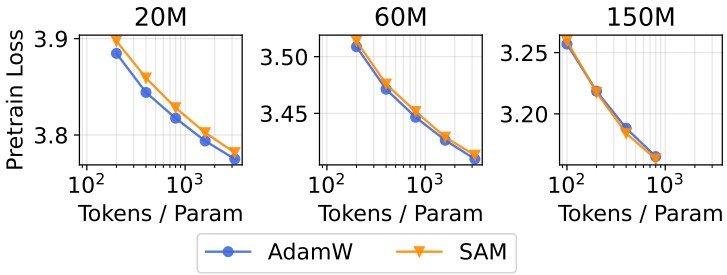

Figure 5: **SAM has worse pre-training loss than AdamW at small scale, but converges as model size increases.** Across model sizes and token budgets, SAM pre-trained models achieve a worse or similar pre-training loss compared to AdamW. However, better pretrain loss alone does not translate into mitigating forgetting.

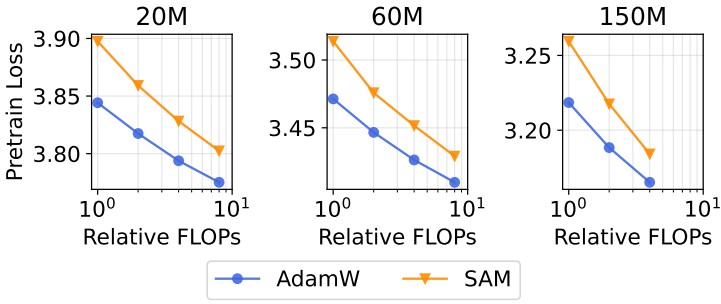

Figure 6: **SAM vs AdamW Pretrain Loss across model sizes in compute matched settings**

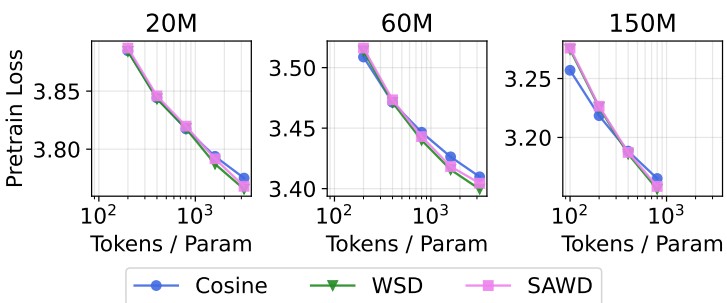

Figure 7: **WSD vs Cosine Pretrain Loss across model sizes**

## D.2 LEARNING-FORGETTING PARETO

### D.2.1 ACROSS DATASETS

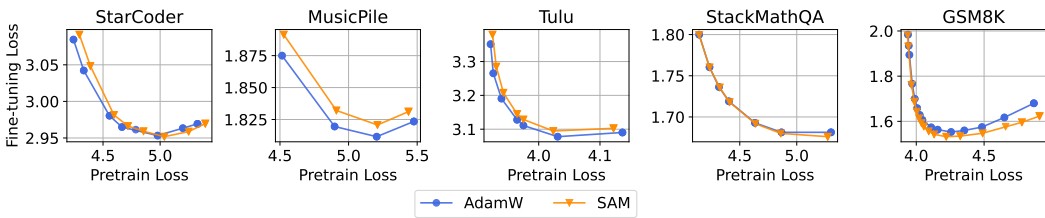

Figure 8: **AdamW vs SAM Learning-Forgetting frontier for OLMo-20M across datasets at 4B tokens**

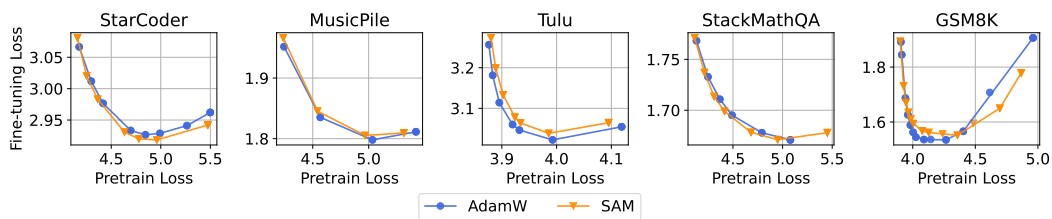

Figure 9: **AdamW vs SAM Learning-Forgetting frontier for OLMo-20M across datasets at 8B tokens**

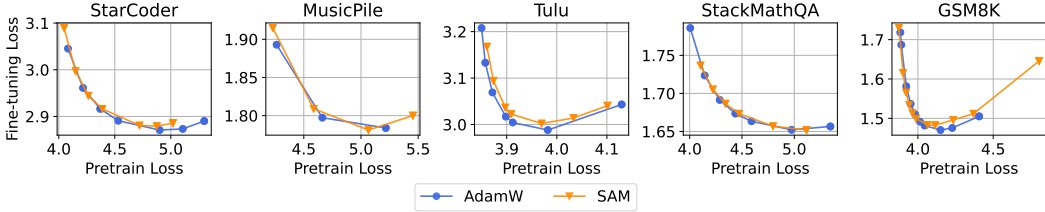

Figure 10: **AdamW vs SAM Learning-Forgetting frontier for OLMo-20M across datasets at 16B tokens**

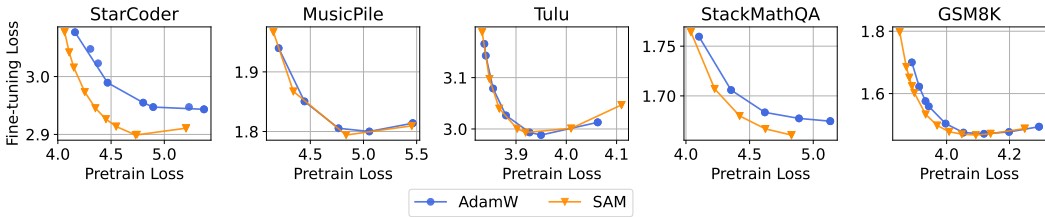

Figure 11: **AdamW vs SAM Learning-Forgetting frontier for OLMo-20M across datasets at 32B tokens**

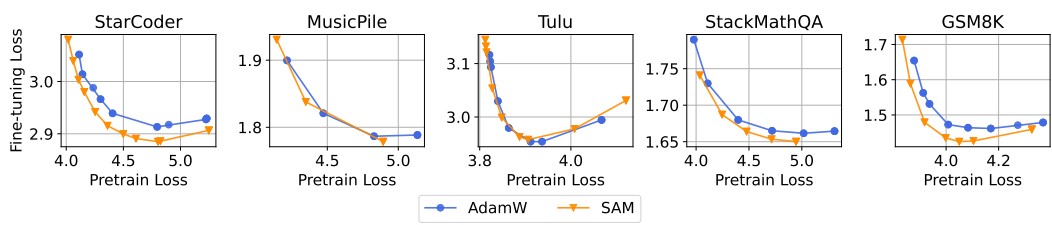

Figure 12: **AdamW vs SAM Learning-Forgetting frontier for OLMo-20M across datasets at 64B tokens**

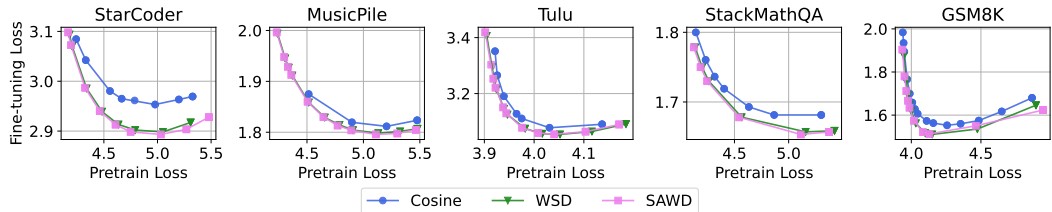

Figure 13: **Cosine vs WSD Learning-Forgetting frontier for OLMo-20M across datasets at 4B tokens**

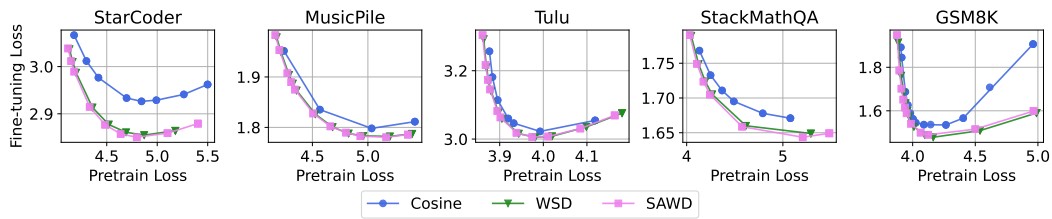

Figure 14: **Cosine vs WSD Learning-Forgetting frontier for OLMo-20M across datasets at 8B tokens**

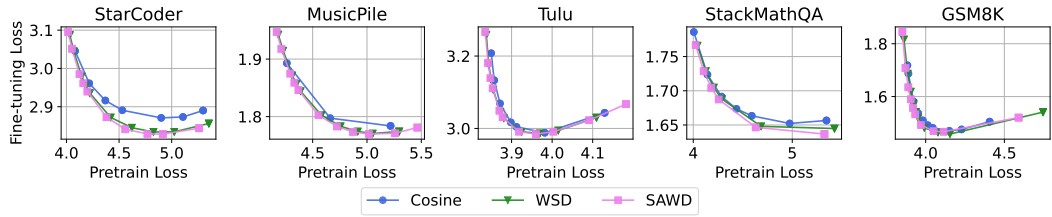

Figure 15: **Cosine vs WSD Learning-Forgetting frontier for OLMo-20M across datasets at 16B tokens**

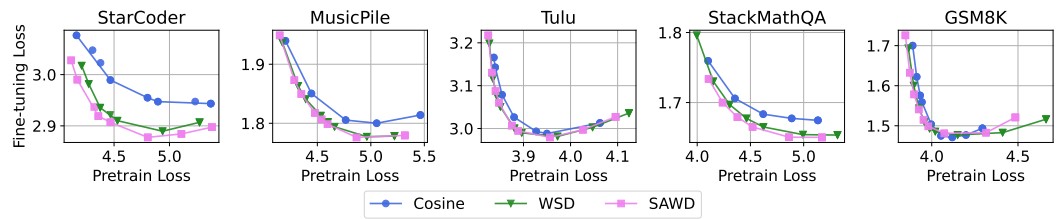

Figure 16: **Cosine vs WSD Learning-Forgetting frontier for OLMo-20M across datasets at 32B tokens**

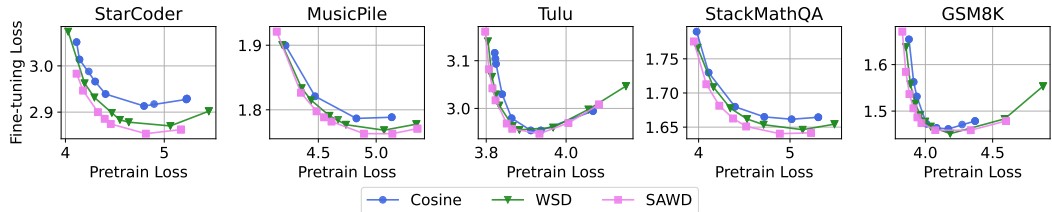

Figure 17: **Cosine vs WSD Learning-Forgetting frontier for OLMo-20M across datasets at 64B tokens**

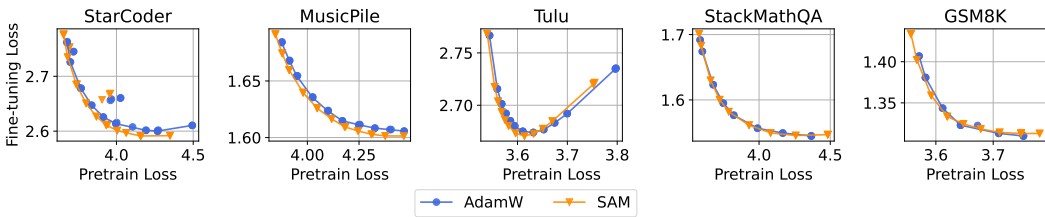

Figure 18: **AdamW vs SAM Learning-Forgetting frontier for OLMo-60M across datasets at 12B tokens**

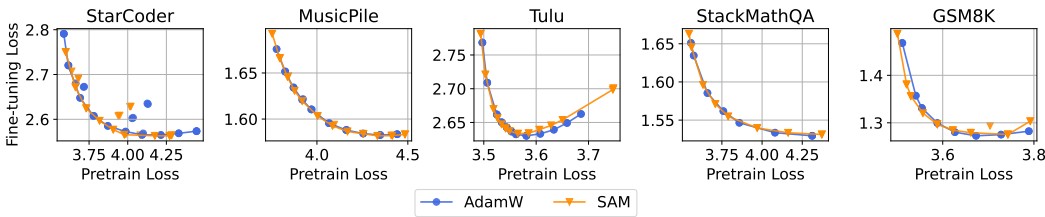

Figure 19: **AdamW vs SAM Learning-Forgetting frontier for OLMo-60M across datasets at 24B tokens**

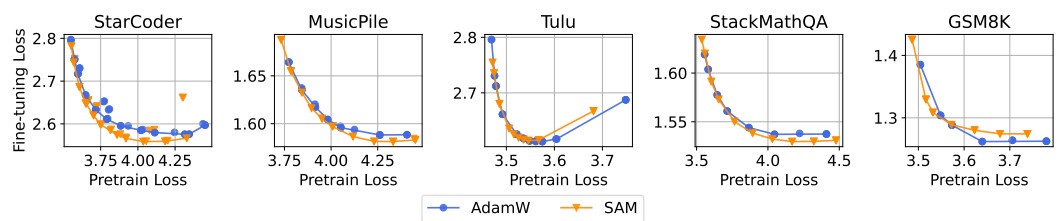

Figure 20: **AdamW vs SAM Learning-Forgetting frontier for OLMo-60M across datasets at 48B tokens**

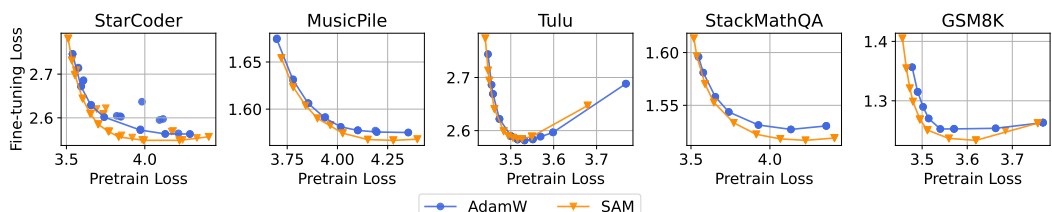

Figure 21: **AdamW vs SAM Learning-Forgetting frontier for OLMo-60M across datasets at 96B tokens**

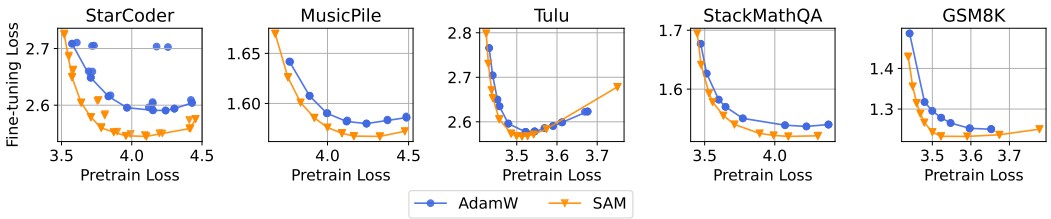

Figure 22: **AdamW vs SAM Learning-Forgetting frontier for OLMo-60M across datasets at 192B tokens**

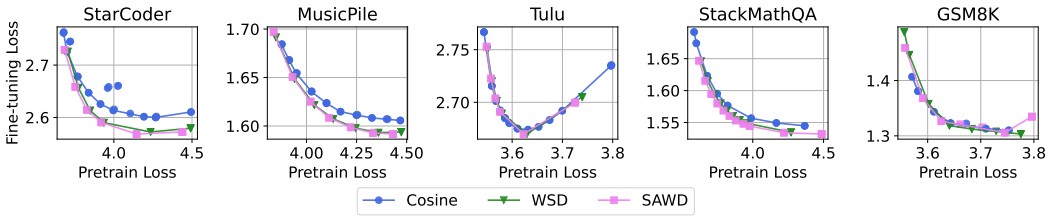

Figure 23: **Cosine vs WSD Learning-Forgetting frontier for OLMo-60M across datasets at 12B tokens**

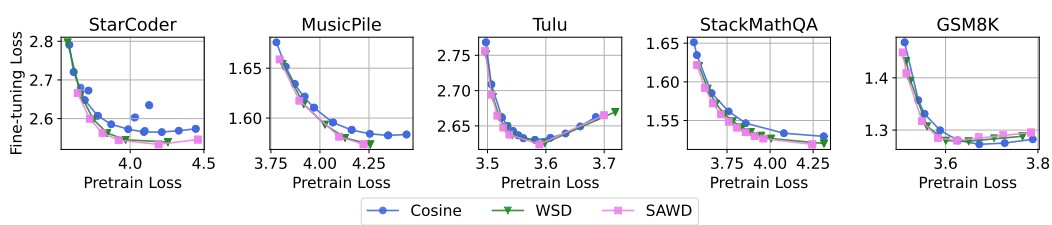

Figure 24: **Cosine vs WSD Learning-Forgetting frontier for OLMo-60M across datasets at 24B tokens**

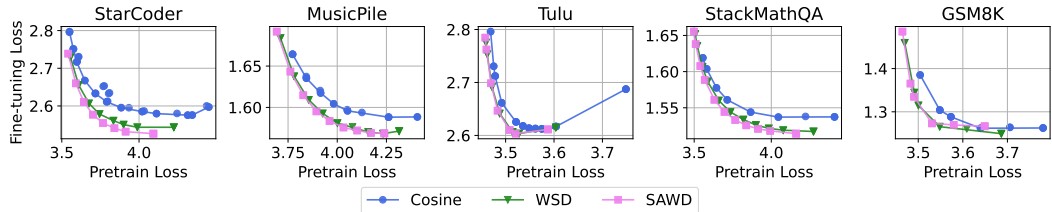

Figure 25: **Cosine vs WSD Learning-Forgetting frontier for OLMo-60M across datasets at 48B tokens**

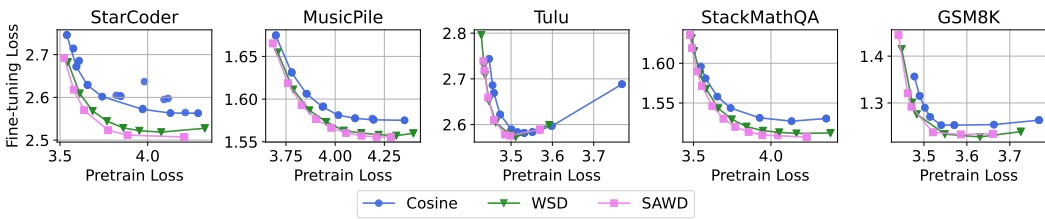

Figure 26: **Cosine vs WSD Learning-Forgetting frontier for OLMo-60M across datasets at 96B tokens**

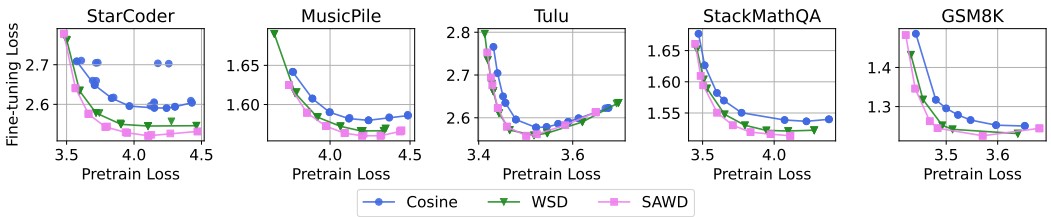

Figure 27: **Cosine vs WSD Learning-Forgetting frontier for OLMo-60M across datasets at 192B tokens**

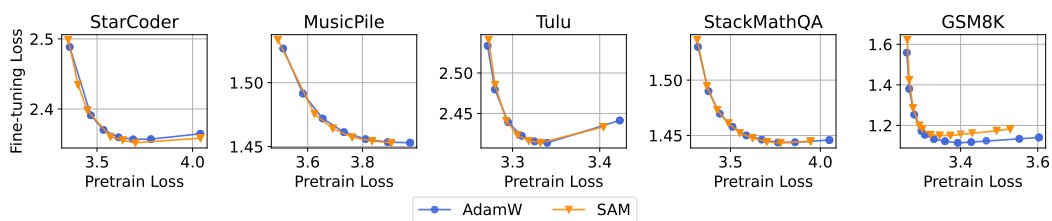

Figure 28: **AdamW vs SAM Learning-Forgetting frontier for OLMo-150M across datasets at 15B tokens**

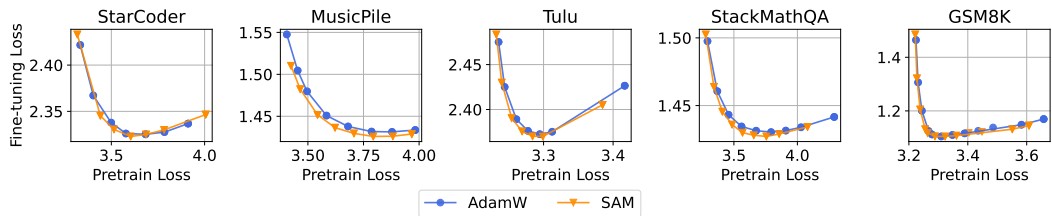

Figure 29: **AdamW vs SAM Learning-Forgetting frontier for OLMo-150M across datasets at 30B tokens**

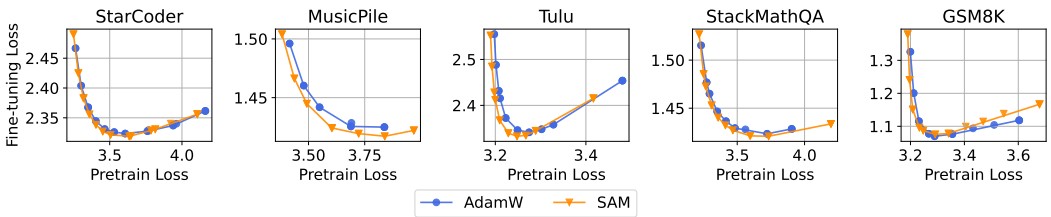

Figure 30: **AdamW vs SAM Learning-Forgetting frontier for OLMo-150M across datasets at 60B tokens**

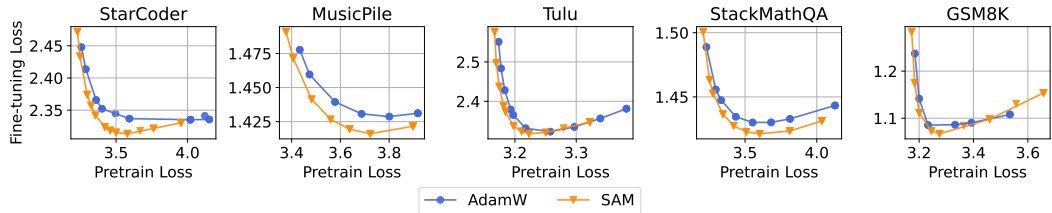

Figure 31: **AdamW vs SAM Learning-Forgetting frontier for OLMo-150M across datasets at 120B tokens**

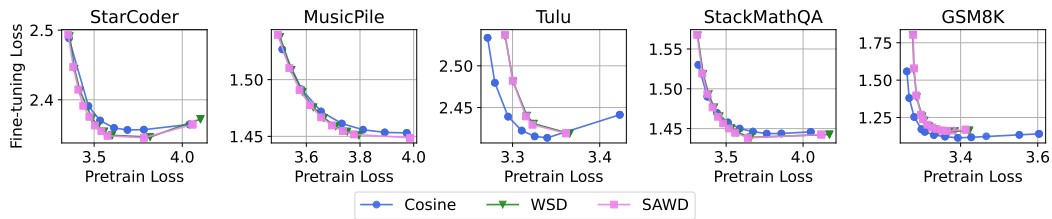

Figure 32: **Cosine vs WSD Learning-Forgetting frontier for OLMo-150M across datasets at 15B tokens**

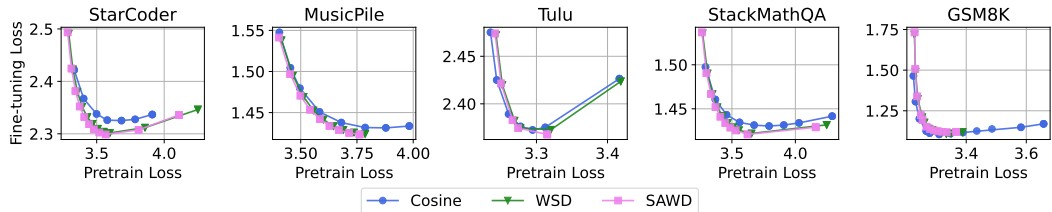

Figure 33: **Cosine vs WSD Learning-Forgetting frontier for OLMo-150M across datasets at 30B tokens**

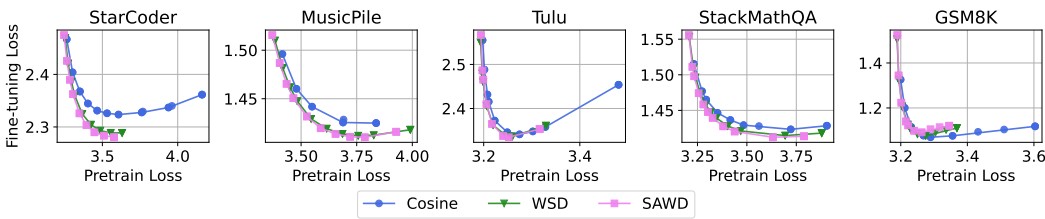

Figure 34: **Cosine vs WSD Learning-Forgetting frontier for OLMo-150M across datasets at 60B tokens**

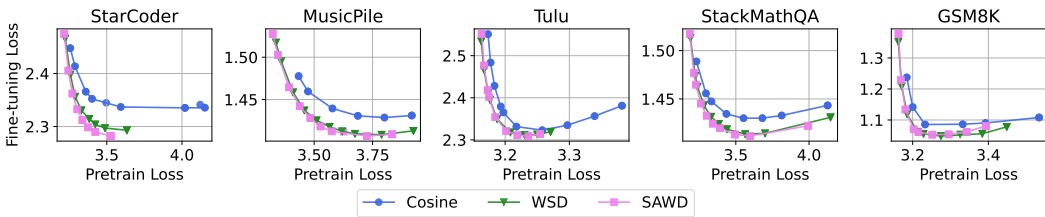

Figure 35: **Cosine vs WSD Learning-Forgetting frontier for OLMo-150M across datasets at 120B tokens**

### D.2.2 SCALING TOKENS

We investigate whether benefits persist as we scale data and model size. In this section, we compare learning–forgetting frontiers for all pretrained checkpoints trained using different configurations of optimizers (AdamW, SAM), learning rate schedule (Cosine, WSD), and model sizes (20M, 60M, 150M) in a token-matched setting. The performance gap between SAM and AdamW *widens* as tokens increase. While Springer et al. (2025) report that AdamW models become increasingly brittle with prolonged training, SAM mitigates this degradation, demonstrating its strongest advantage in high-token regimes. We further see similar trends when comparing WSD and Cosine schedules.

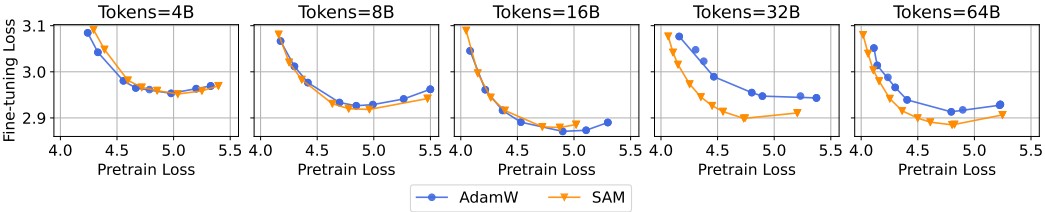

Figure 36: **AdamW vs SAM Starcoder 20M**

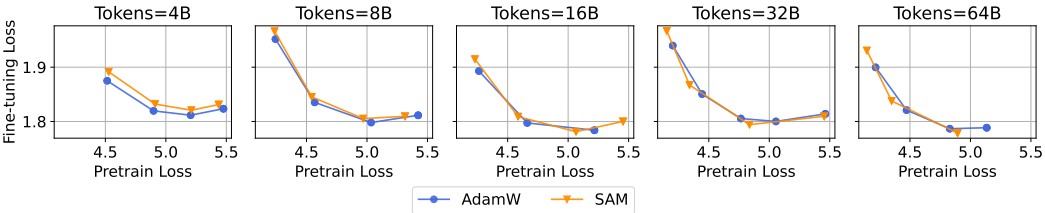

Figure 37: **AdamW vs SAM Musicpile 20M**

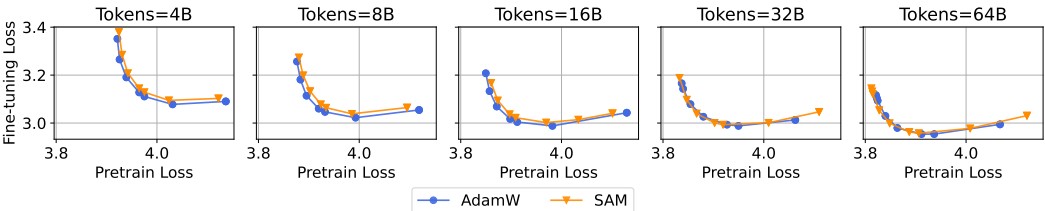

Figure 38: **AdamW vs SAM Tulu 20M**

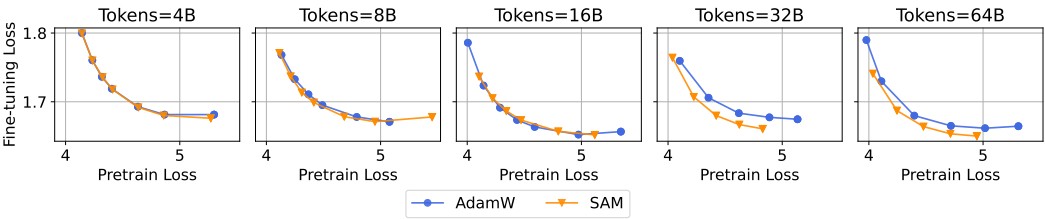

Figure 39: **AdamW vs SAM StackmathQA 20M**

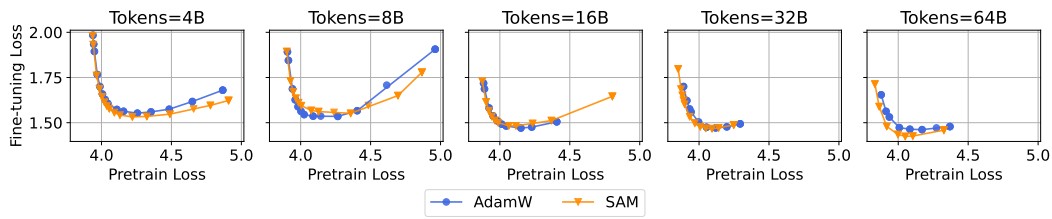

Figure 40: **AdamW vs SAM GSM8K 20M**

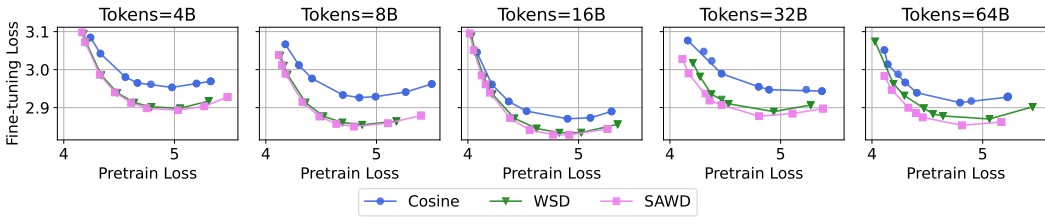

Figure 41: **WSD vs Cosine Starcoder 20M**

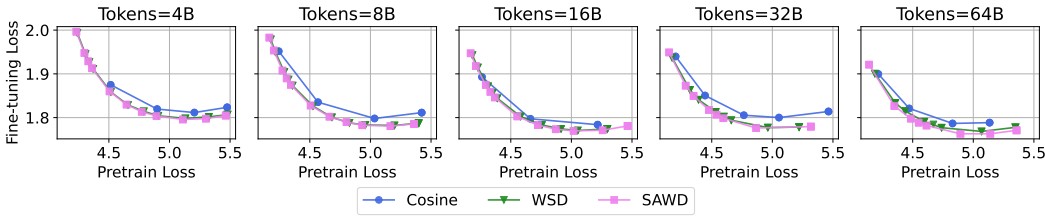

Figure 42: **WSD vs Cosine Musicpile 20M**

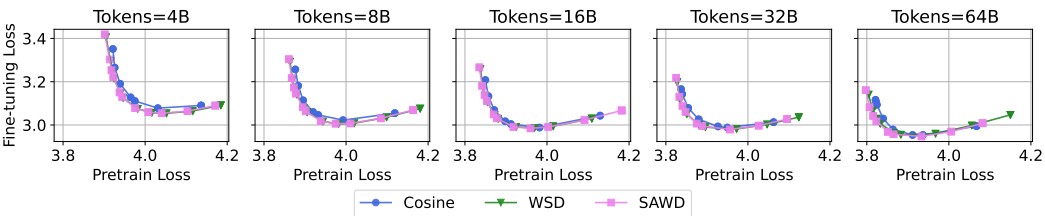

Figure 43: **WSD vs Cosine Tulu 20M**

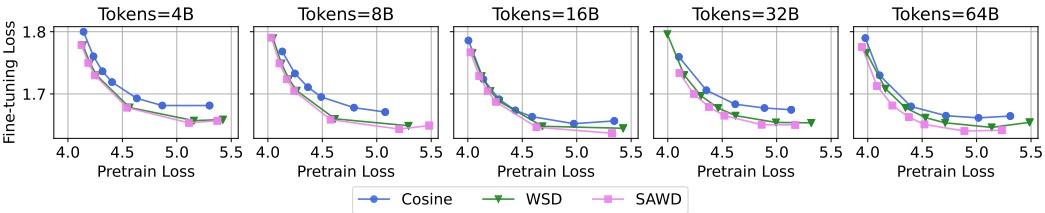

Figure 44: **WSD vs Cosine StackmathQA 20M**

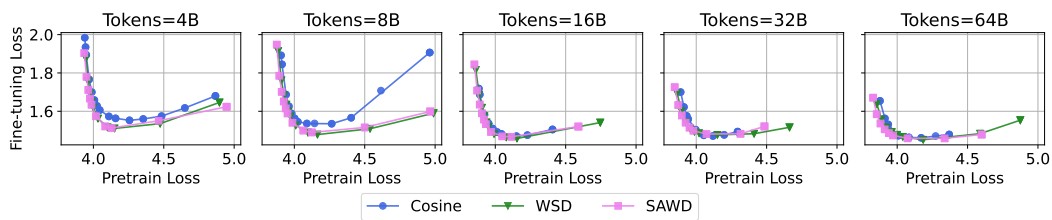

Figure 45: **WSD vs Cosine GSM8K 20M**

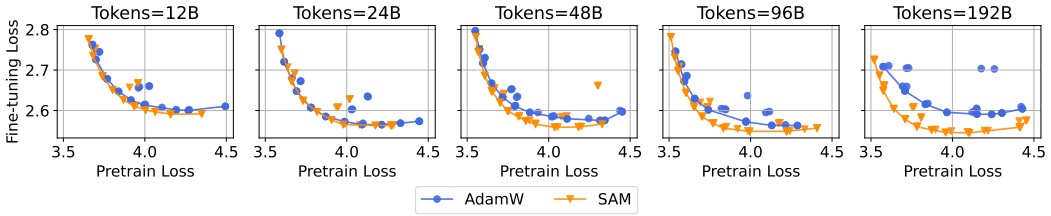

Figure 46: **AdamW vs SAM Starcoder 60M**

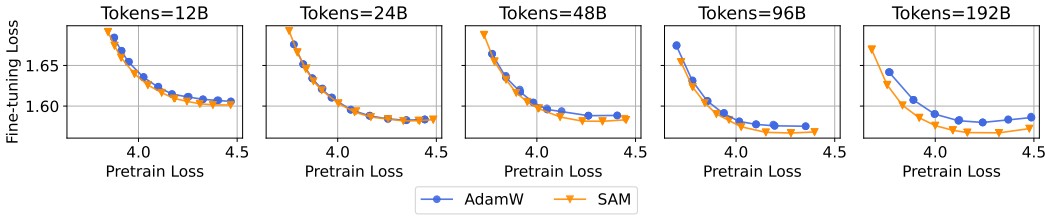

Figure 47: **AdamW vs SAM Musicpile 60M**

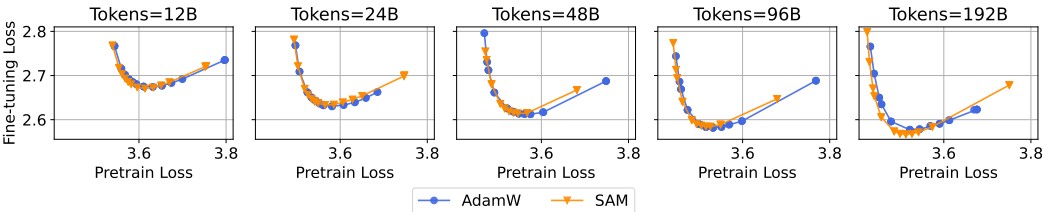

Figure 48: **AdamW vs SAM Tulu 60M**

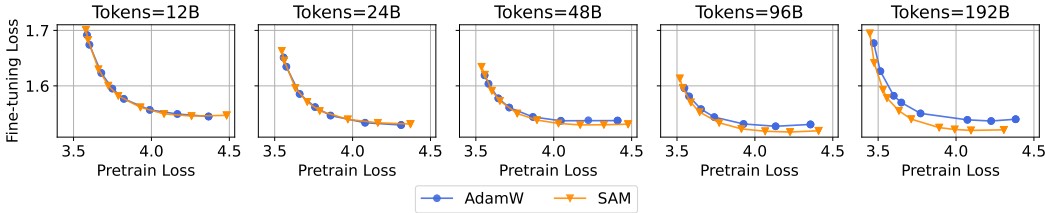

Figure 49: **AdamW vs SAM StackmathQA 60M**

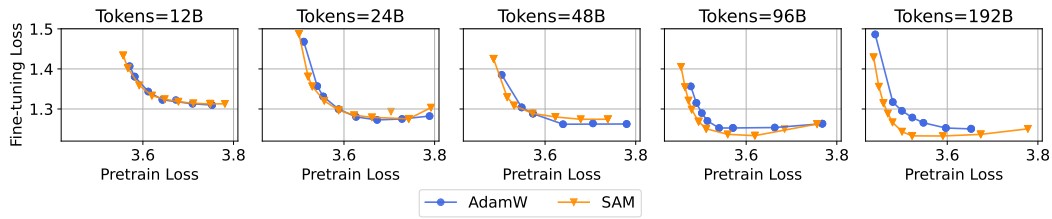

Figure 50: **AdamW vs SAM GSM8K 60M**

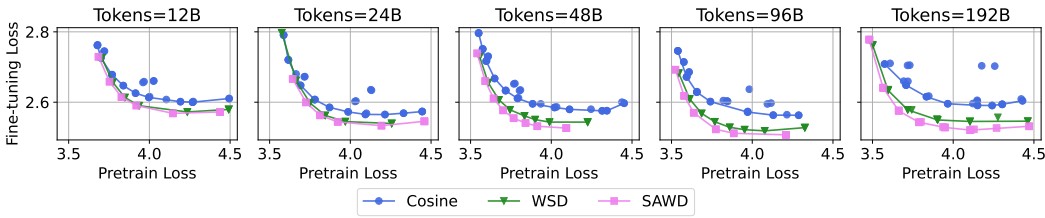

Figure 51: **WSD vs Cosine Starcoder 60M**

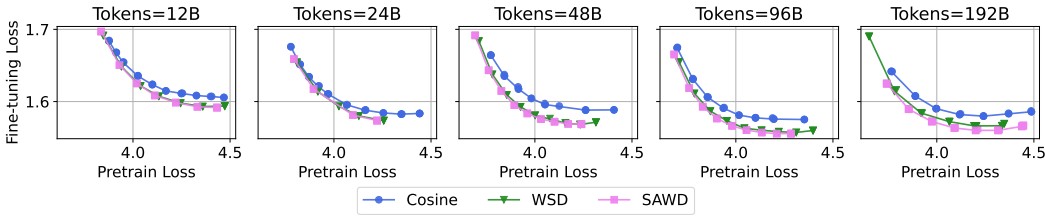

Figure 52: **WSD vs Cosine Musicpile 60M**

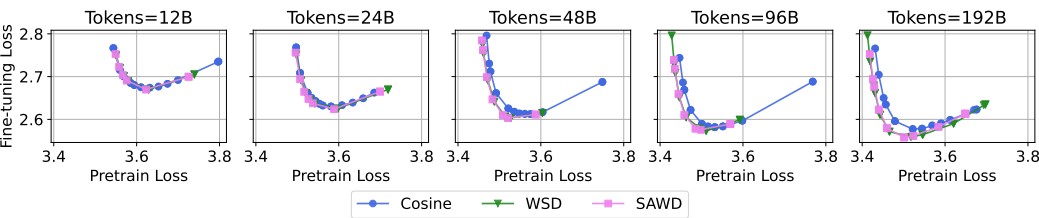

Figure 53: **WSD vs Cosine Tulu 60M**

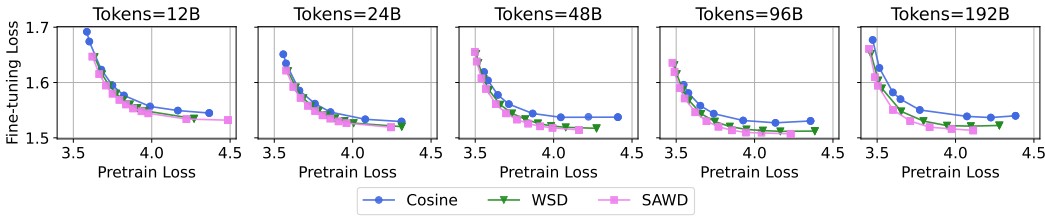

Figure 54: **WSD vs Cosine StackmathQA 60M**

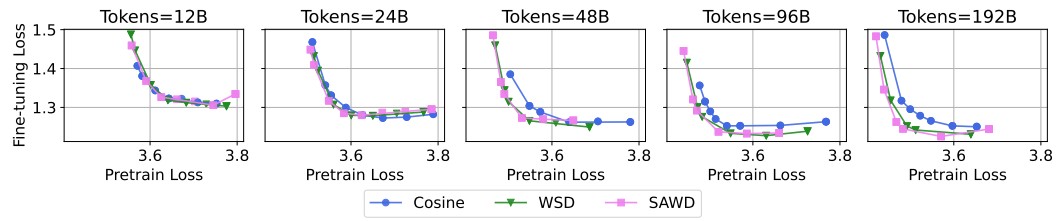

Figure 55: **WSD vs Cosine GSM8K 60M**

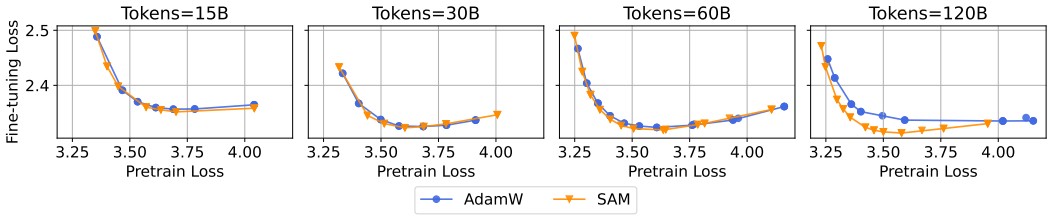

Figure 56: **AdamW vs SAM Starcoder 150M**

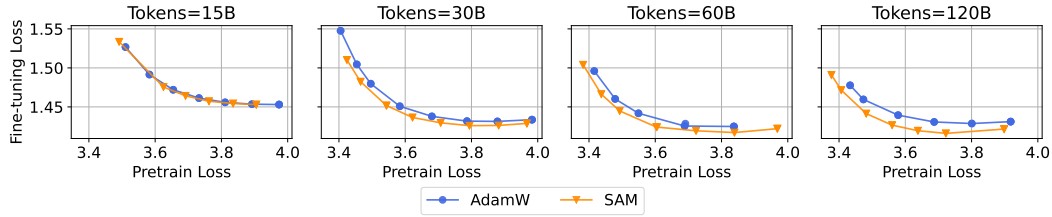

Figure 57: **AdamW vs SAM Musicpile 150M**

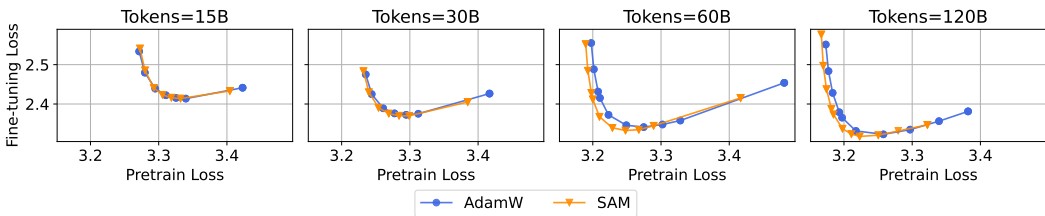

Figure 58: **AdamW vs SAM Tulu 150M**

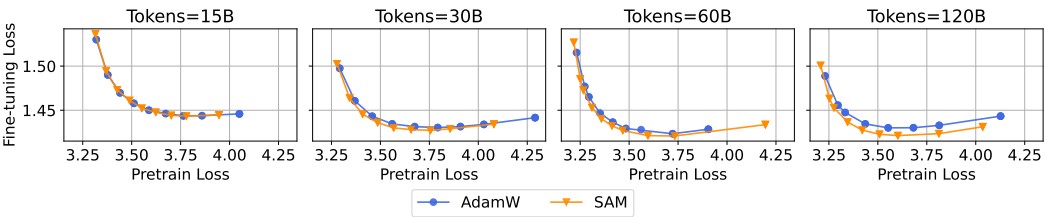

Figure 59: **AdamW vs SAM StackmathQA 150M**

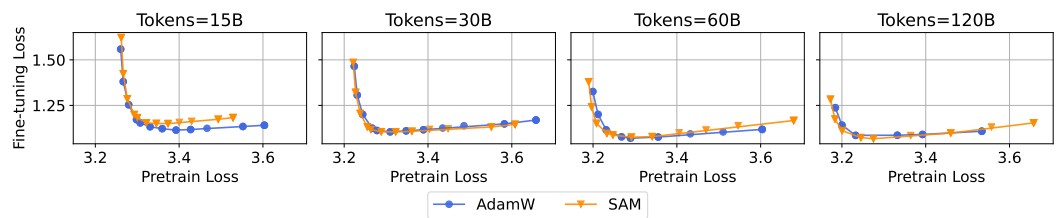

Figure 60: **AdamW vs SAM GSM8K 150M**

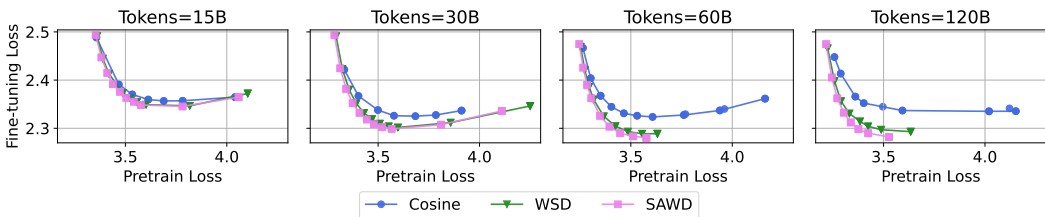

Figure 61: **WSD vs Cosine Starcoder 150M**

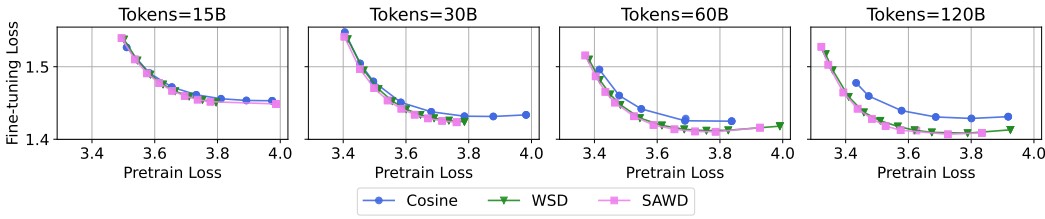

Figure 62: **WSD vs Cosine Musicpile 150M**

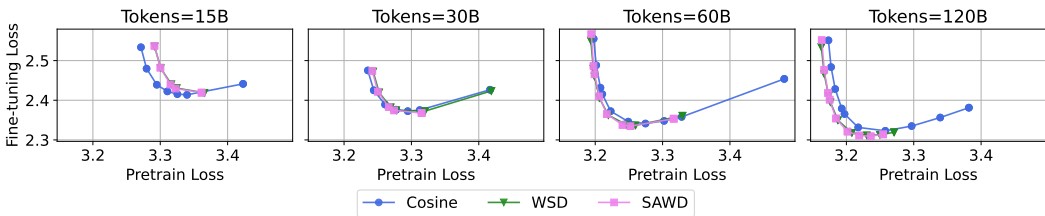

Figure 63: **WSD vs Cosine Tulu 150M**

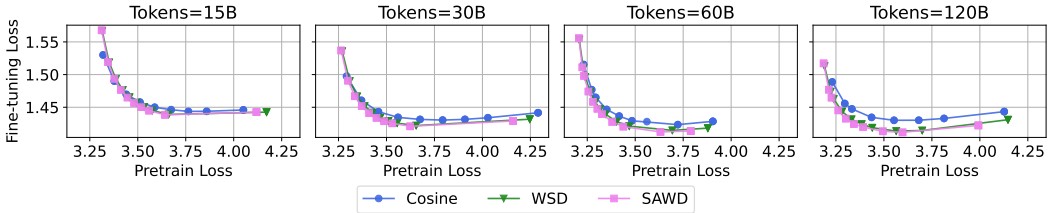

Figure 64: **WSD vs Cosine StackmathQA 150M**

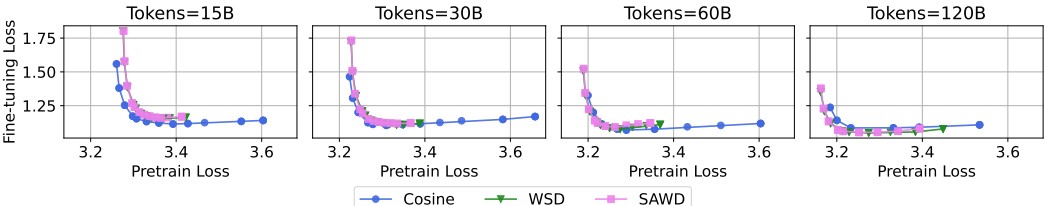

Figure 65: **WSD vs Cosine GSM8K 150M**

### D.2.3   COMPUTE MATCHED

Since SAM's two-step update doubles per-iteration compute cost, we also compare SAM and AdamW in a compute-matched setting, where the SAM models are trained on half the tokens to equalize total FLOPs. Even under this stricter comparison, SAM often maintains competitive or superior learning–forgetting frontiers, particularly at larger model sizes and higher token-to-parameter ratios.

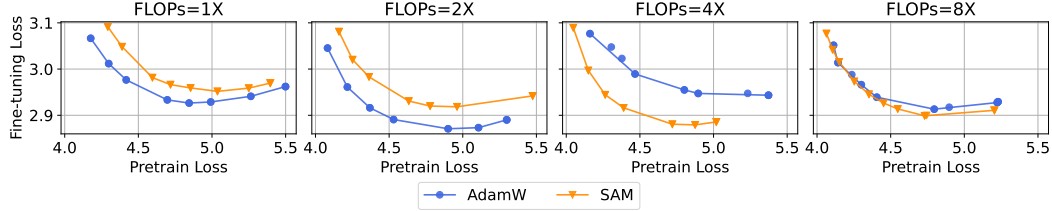

Figure 66: **AdamW vs SAM Starcoder 20M (Compute Matched)**

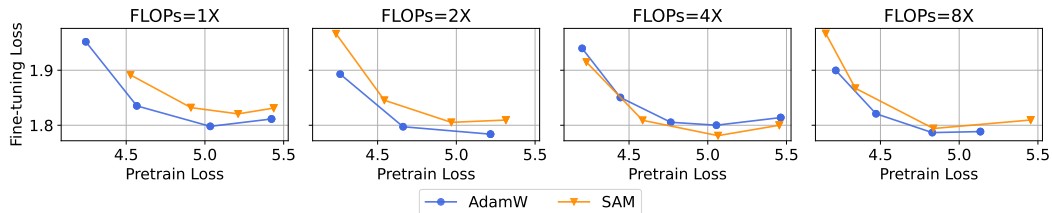

Figure 67: **AdamW vs SAM Musicpile 20M (Compute Matched)**

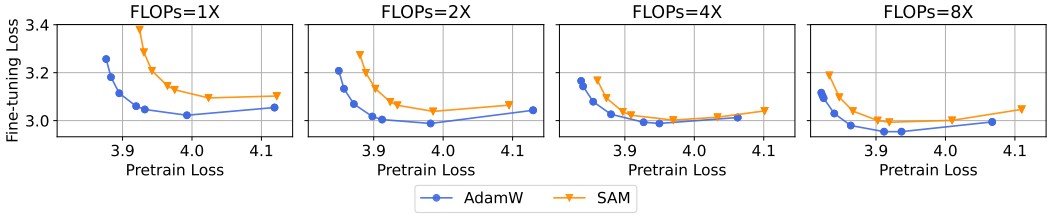

Figure 68: **AdamW vs SAM Tulu 20M (Compute Matched)**

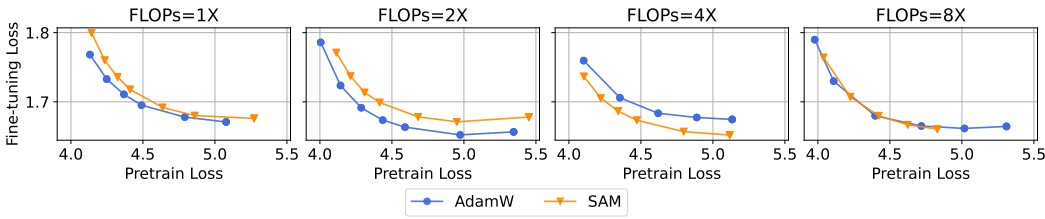

Figure 69: **AdamW vs SAM StackmathQA 20M (Compute Matched)**

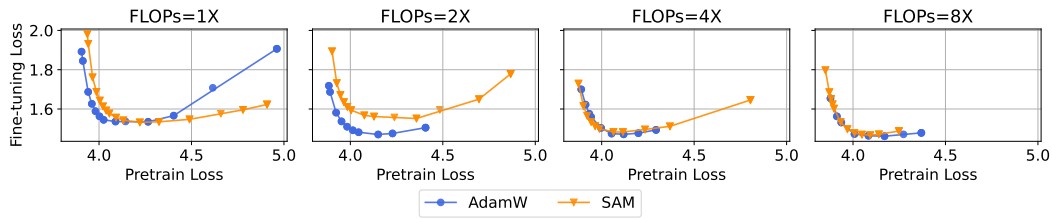

Figure 70: **AdamW vs SAM GSM8K 20M (Compute Matched)**

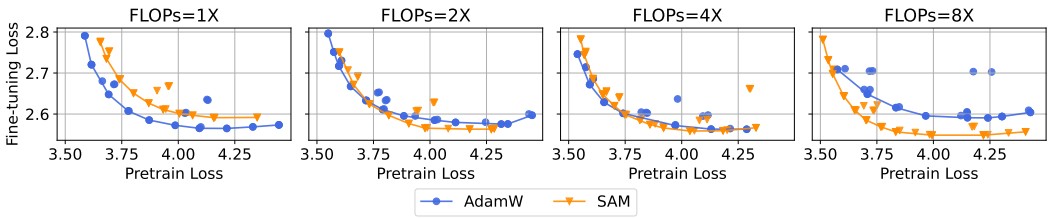

Figure 71: **AdamW vs SAM Starcoder 60M (Compute Matched)**

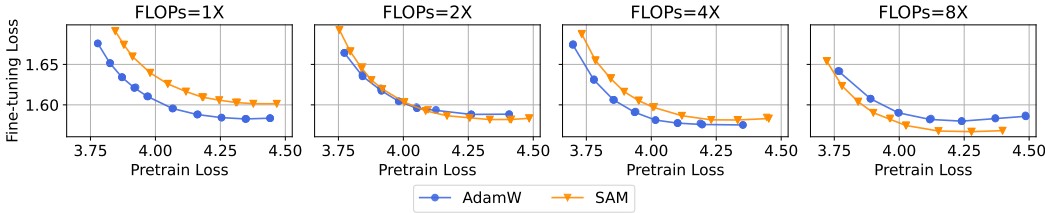

Figure 72: **AdamW vs SAM Musicpile 60M (Compute Matched)**

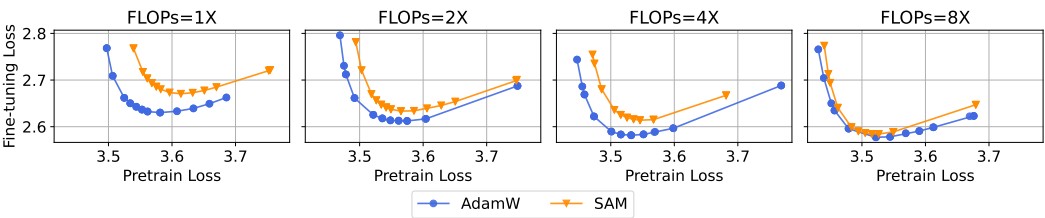

Figure 73: **AdamW vs SAM Tulu 60M (Compute Matched)**

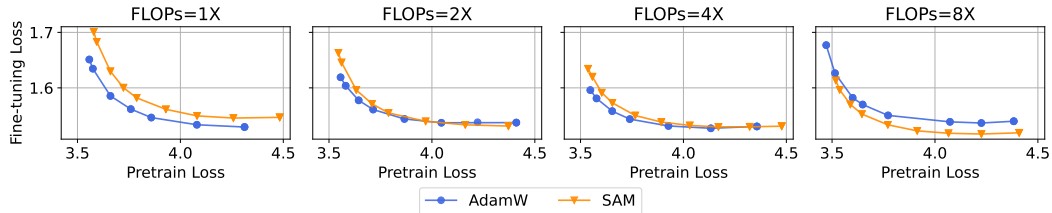

Figure 74: **AdamW vs SAM StackmathQA 60M (Compute Matched)**

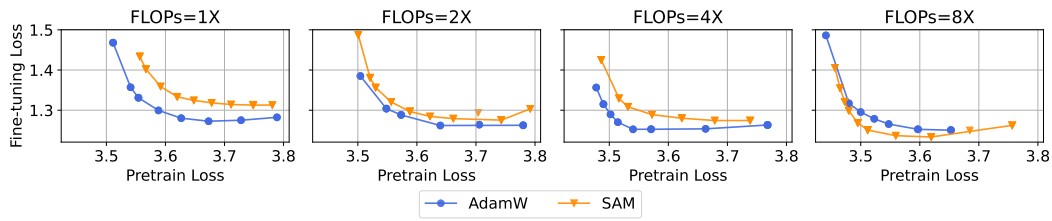

Figure 75: **AdamW vs SAM GSM8K 60M (Compute Matched)**

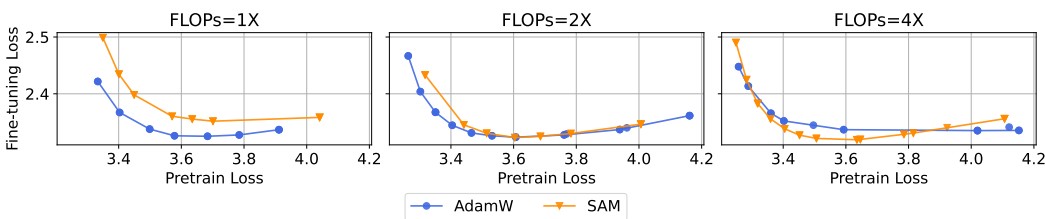

Figure 76: **AdamW vs SAM Starcoder 150M (Compute Matched)**

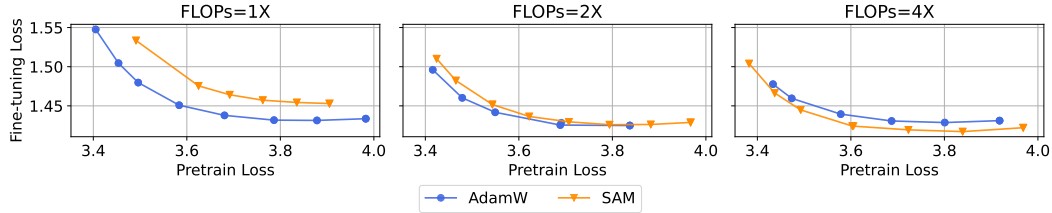

Figure 77: **AdamW vs SAM Musicpile 150M (Compute Matched)**

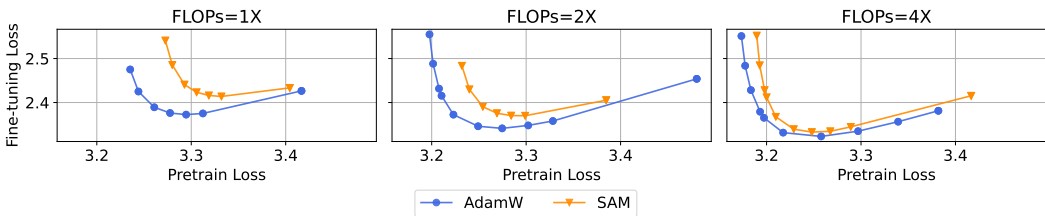

Figure 78: **AdamW vs SAM Tulu 150M (Compute Matched)**

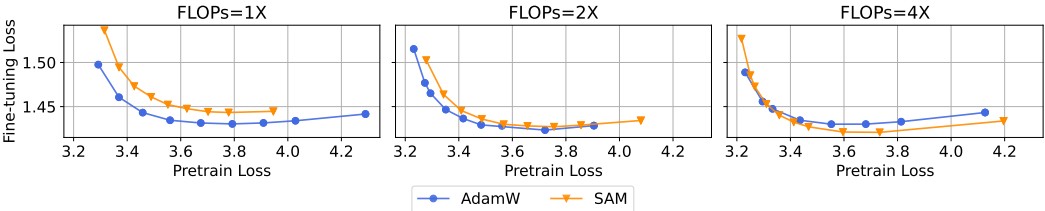

Figure 79: **AdamW vs SAM StackmathQA 150M (Compute Matched)**

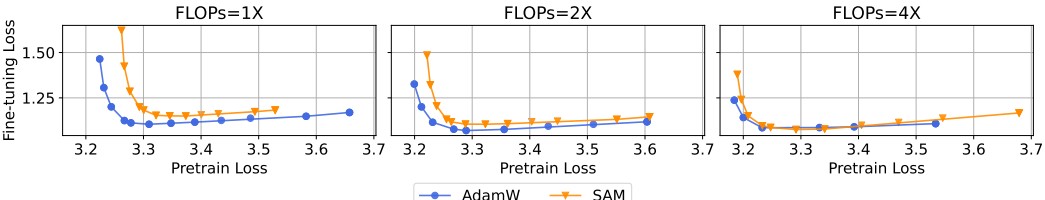

Figure 80: **AdamW vs SAM GSM8K 150M (Compute Matched)**

### D.2.4 SCALING MODEL SIZE

We analyze whether benefits persist as model size increases. In this sction, we plot learning–forgetting frontiers across model sizes at multiple fixed *token-per-parameter* ratios (200, 400, 800). The performance gap between SAM and AdamW remains consistent and often *widens* at larger scales, suggesting benefits do not diminish but become more pronounced.

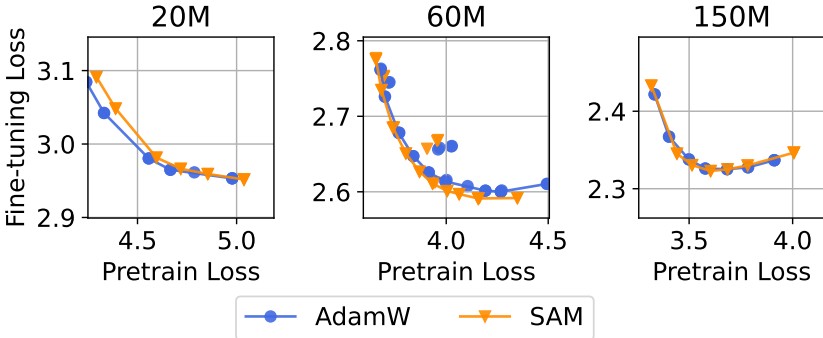

Figure 81: **AdamW vs SAM Learning forgetting frontier across Model Sizes at 200 token / param for Starcoder**

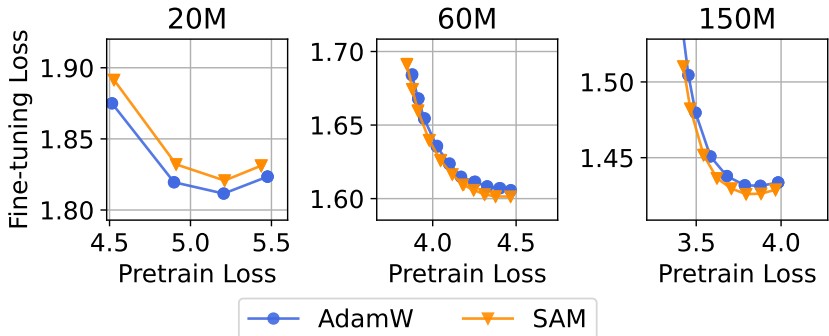

Figure 82: **AdamW vs SAM Learning forgetting frontier across Model Sizes at 200 token / param for Musicpile**

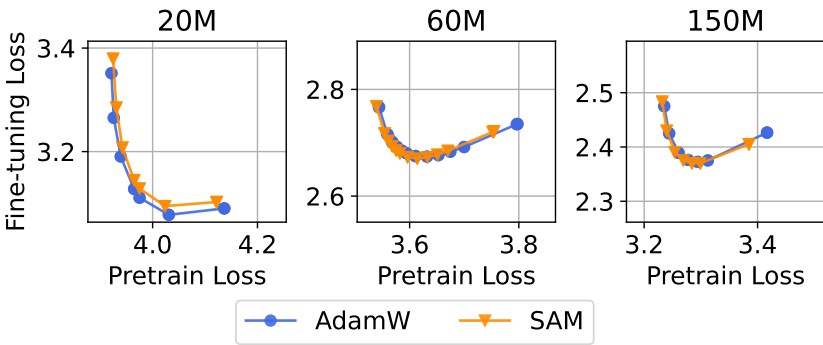

Figure 83: **AdamW vs SAM Learning forgetting frontier across Model Sizes at 200 token / param for Tulu**

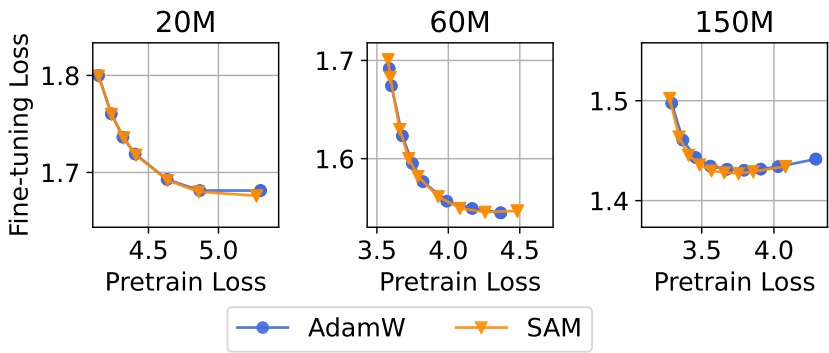

Figure 84: **AdamW vs SAM Learning forgetting frontier across Model Sizes at 200 token / param for StackmathQA**

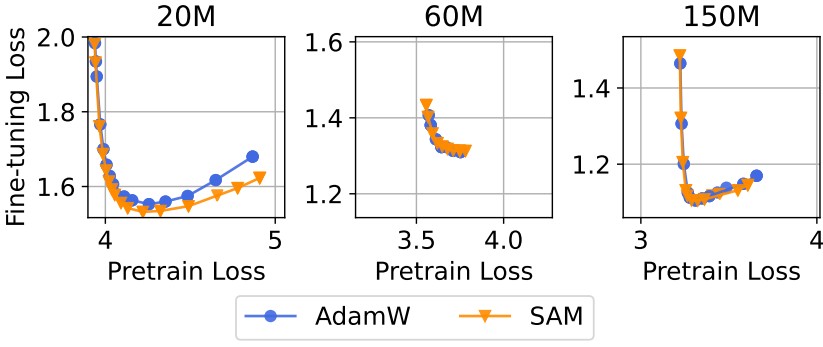

Figure 85: **AdamW vs SAM Learning forgetting frontier across Model Sizes at 200 token / param for GSM8K**

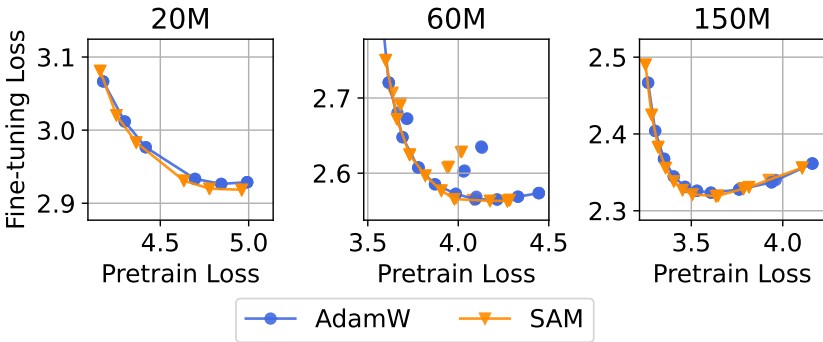

Figure 86: **AdamW vs SAM Learning forgetting frontier across Model Sizes at 400 token / param for Starcoder**

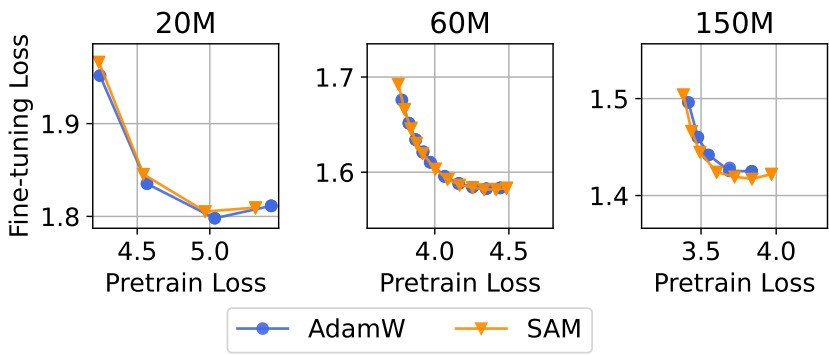

Figure 87: **AdamW vs SAM Learning forgetting frontier across Model Sizes at 400 token / param for Musicpile**

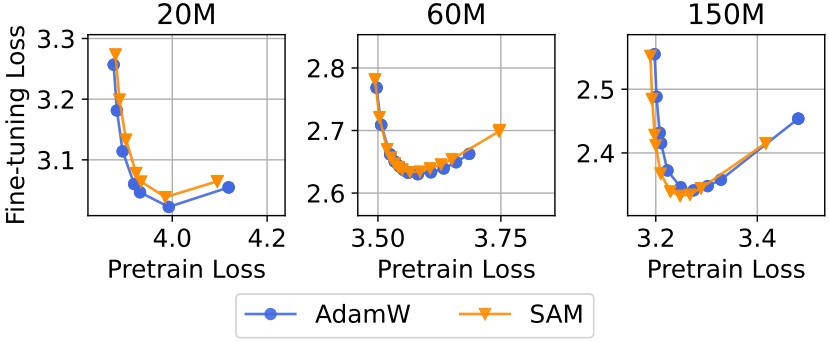

Figure 88: **AdamW vs SAM Learning forgetting frontier across Model Sizes at 400 token / param for Tulu**

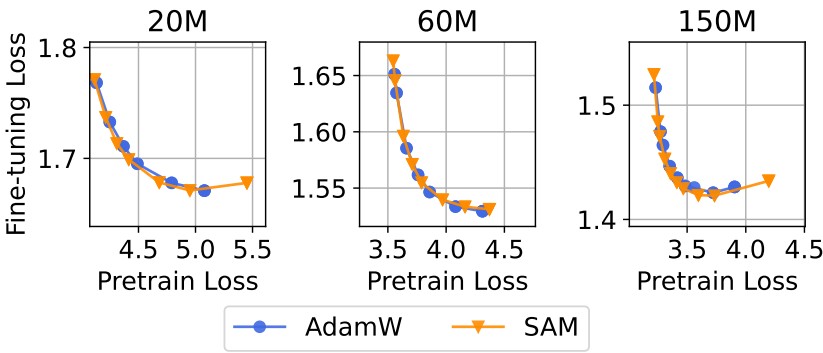

Figure 89: **AdamW vs SAM Learning forgetting frontier across Model Sizes at 400 token / param for StackmathQA**

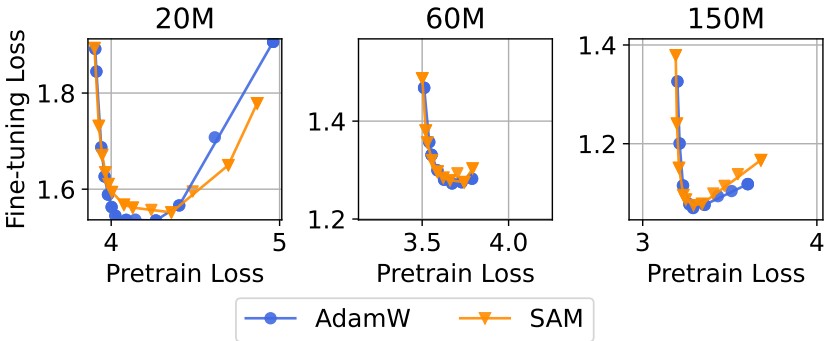

Figure 90: **AdamW vs SAM Learning forgetting frontier across Model Sizes at 400 token / param for GSM8K**

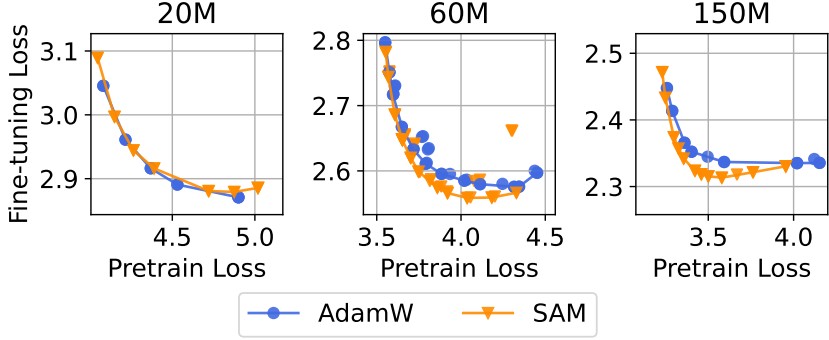

Figure 91: **AdamW vs SAM Learning forgetting frontier across Model Sizes at 800 token / param for Starcoder**

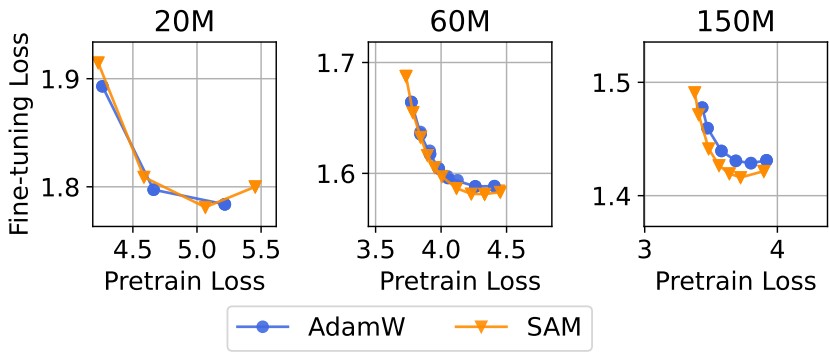

Figure 92: **AdamW vs SAM Learning forgetting frontier across Model Sizes at 800 token / param for Musicpile**

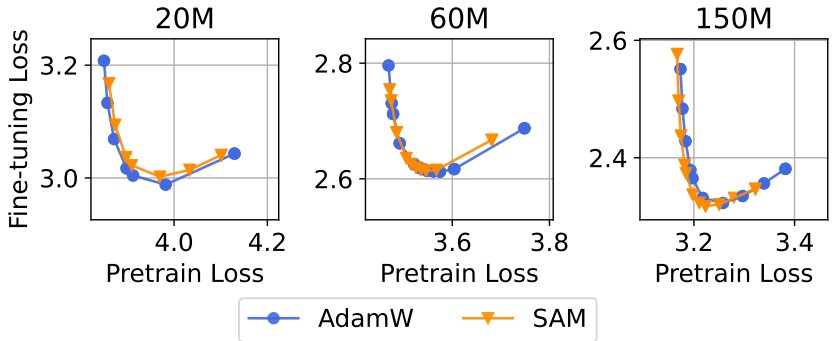

Figure 93: **AdamW vs SAM Learning forgetting frontier across Model Sizes at 800 token / param for Tulu**

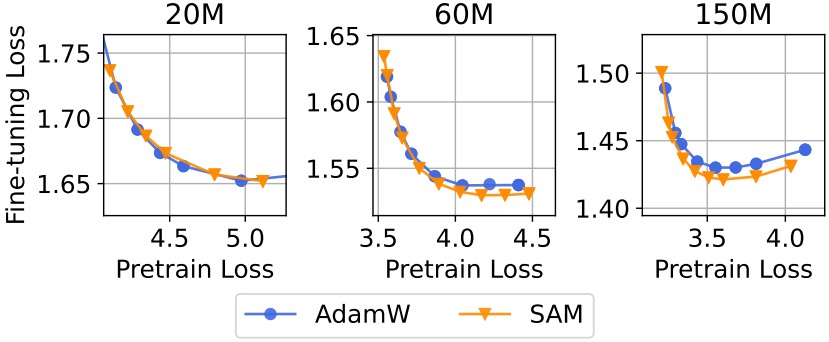

Figure 94: **AdamW vs SAM Learning forgetting frontier across Model Sizes at 800 token / param for StackmathQA**

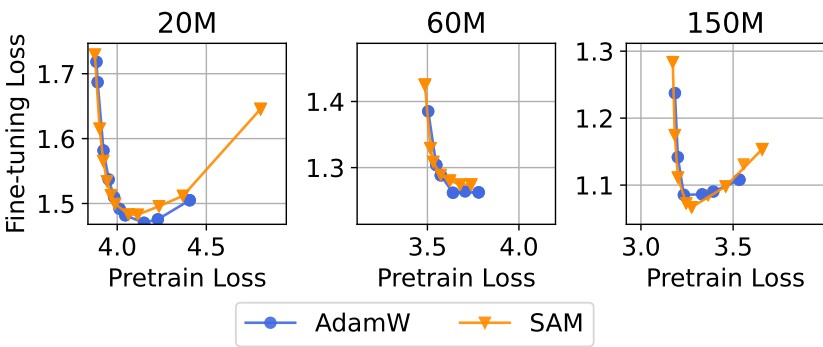

Figure 95: **AdamW vs SAM Learning forgetting frontier across Model Sizes at 800 token / param for GSM8K**

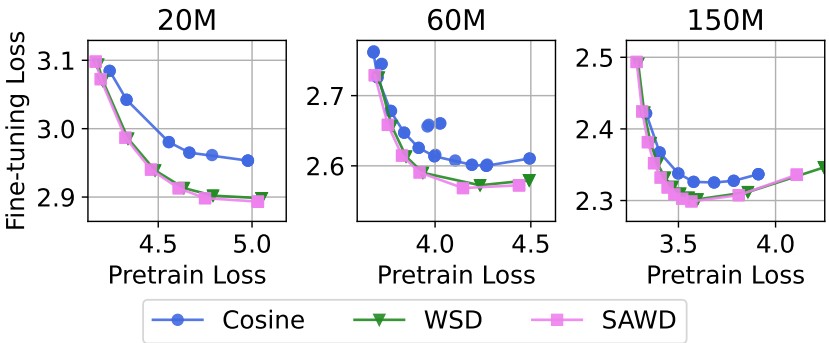

Figure 96: **Cosine vs WSD Learning forgetting frontier across Model Sizes at 200 token / param for Starcoder**

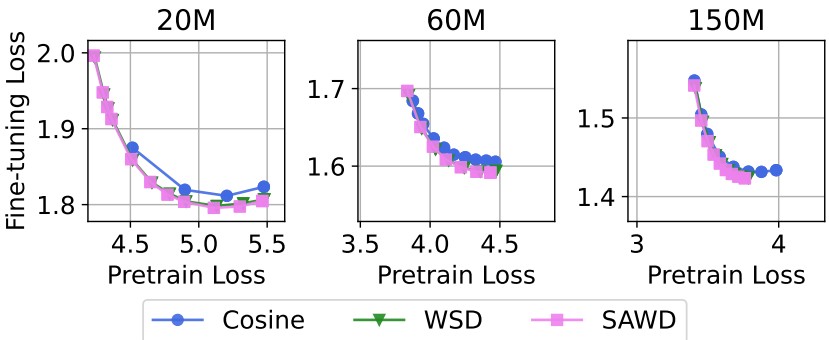

Figure 97: **Cosine vs WSD Learning forgetting frontier across Model Sizes at 200 token / param for Musicpile**

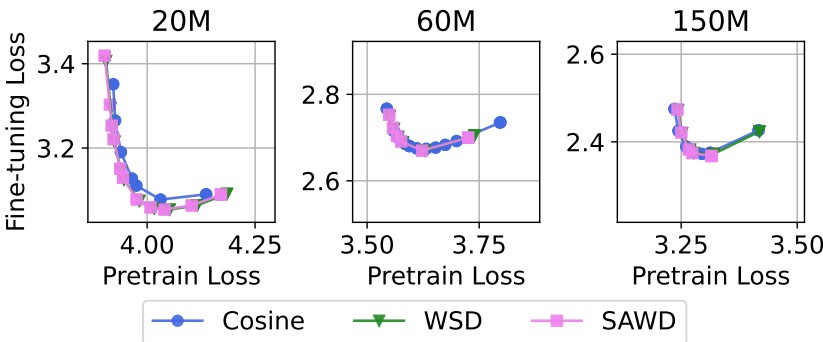

Figure 98: **Cosine vs WSD Learning forgetting frontier across Model Sizes at 200 token / param for Tulu**

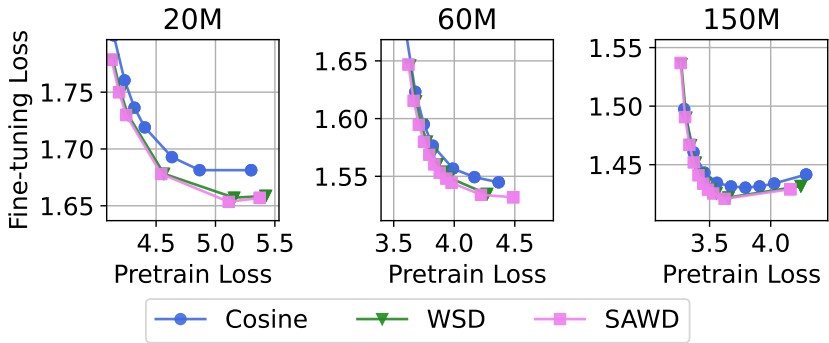

Figure 99: **Cosine vs WSD Learning forgetting frontier across Model Sizes at 200 token / param for StackmathQA**

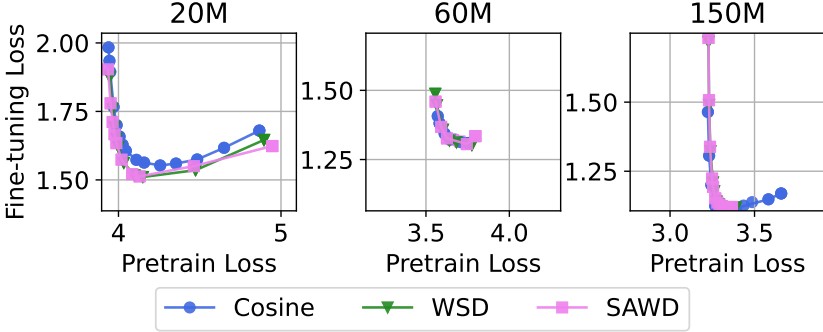

Figure 100: **Cosine vs WSD Learning forgetting frontier across Model Sizes at 200 token / param for GSM8K**

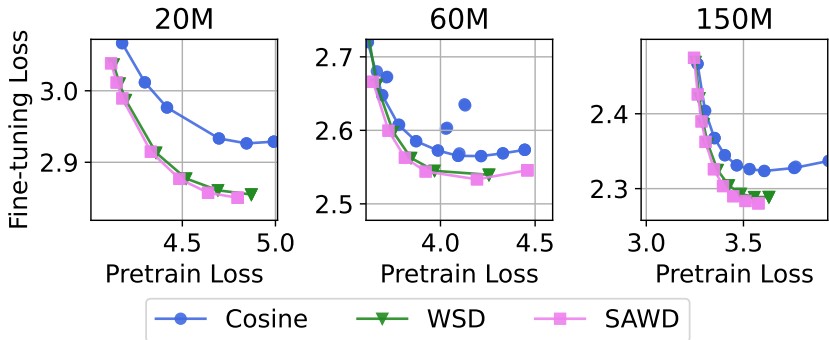

Figure 101: **Cosine vs WSD Learning forgetting frontier across Model Sizes at 400 token / param for Starcoder**

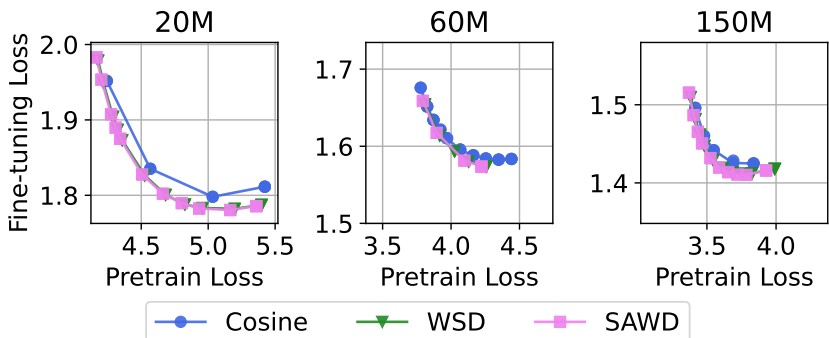

Figure 102: **Cosine vs WSD Learning forgetting frontier across Model Sizes at 400 token / param for Musicpile**

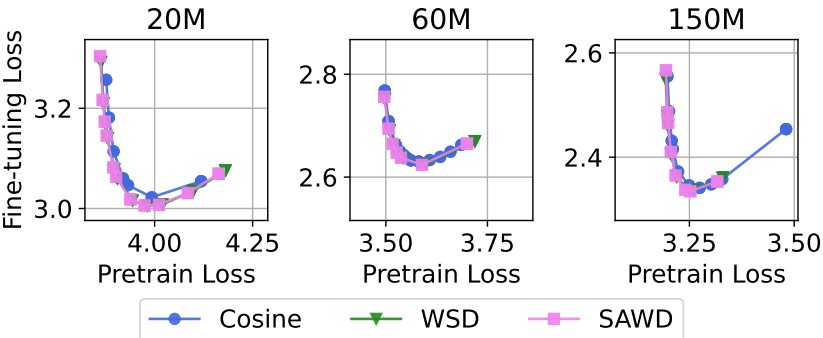

Figure 103: **Cosine vs WSD Learning forgetting frontier across Model Sizes at 400 token / param for Tulu**

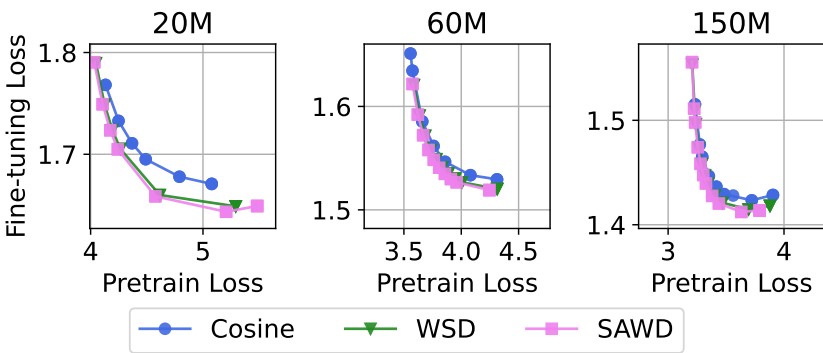

Figure 104: **Cosine vs WSD Learning forgetting frontier across Model Sizes at 400 token / param for StackmathQA**

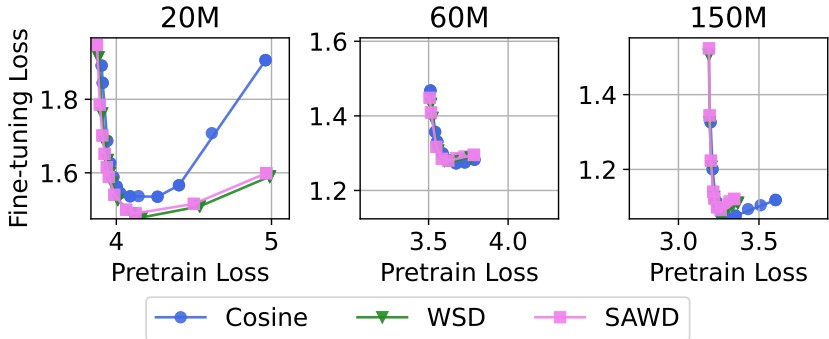

Figure 105: **Cosine vs WSD Learning forgetting frontier across Model Sizes at 400 token / param for GSM8K**

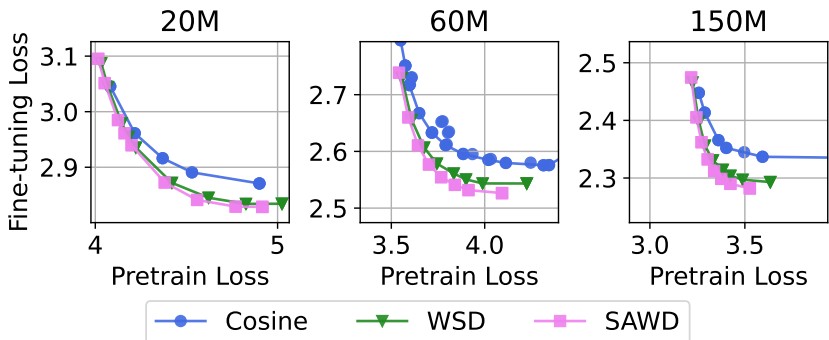

Figure 106: **Cosine vs WSD Learning forgetting frontier across Model Sizes at 800 token / param for Starcoder**

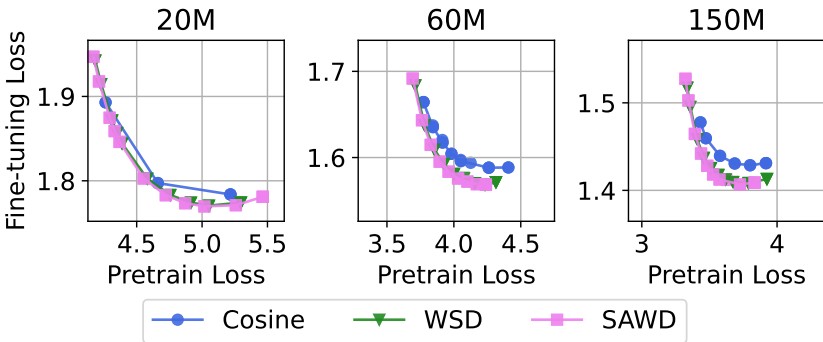

Figure 107: **Cosine vs WSD Learning forgetting frontier across Model Sizes at 800 token / param for Musicpile**

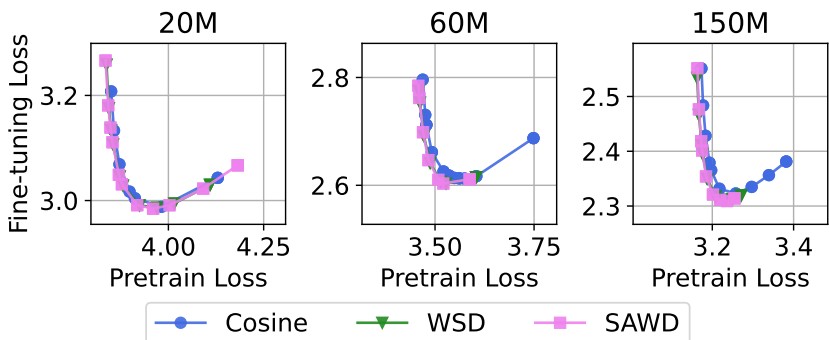

Figure 108: **Cosine vs WSD Learning forgetting frontier across Model Sizes at 800 token / param for Tulu**

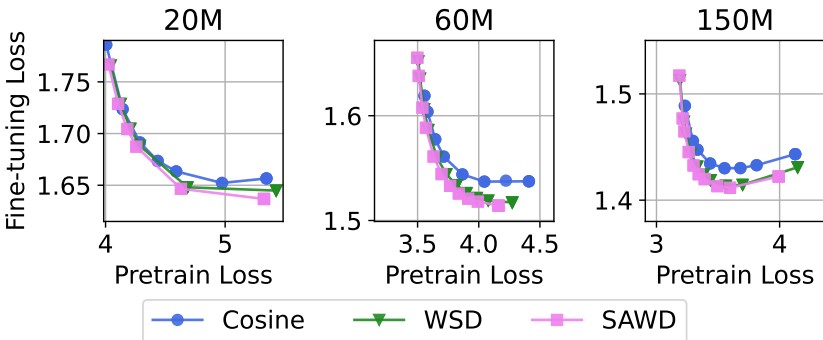

Figure 109: **Cosine vs WSD Learning forgetting frontier across Model Sizes at 800 token / param for StackmathQA**

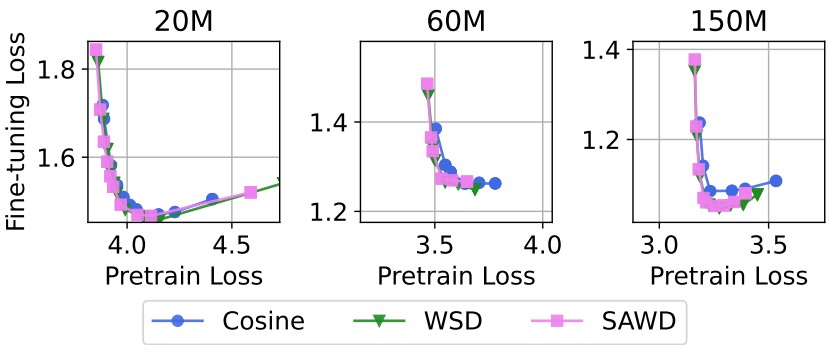

Figure 110: **Cosine vs WSD Learning forgetting frontier across Model Sizes at 800 token / param for GSM8K**

## D.3 LEARNING RATE SCHEDULES

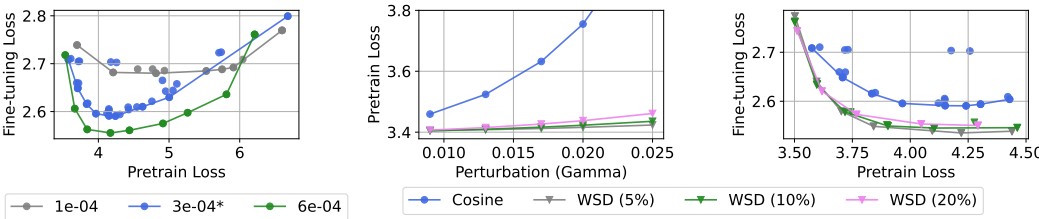

Figure 111: **Learning-rate schedules with shorter decay phases mitigate forgetting.** OLMo-60M models trained for 192B tokens with cosine schedules at varying peak learning rates (left) and WSD schedules with different annealing fractions (middle/right) and fine-tuned on StarCoder. Higher peak learning rates and shorter decay phases yield better learning–forgetting tradeoffs.

## D.4 CATASTROPHIC OVER-TRAINING TRADEOFF

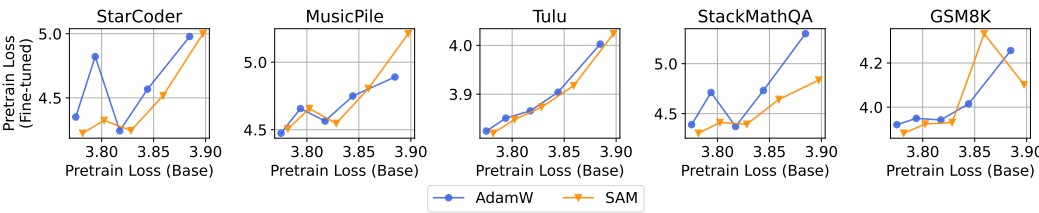

Figure 112: **AdamW vs SAM Catastrophic overtraining tradeoff for OLMo-20M across datasets**

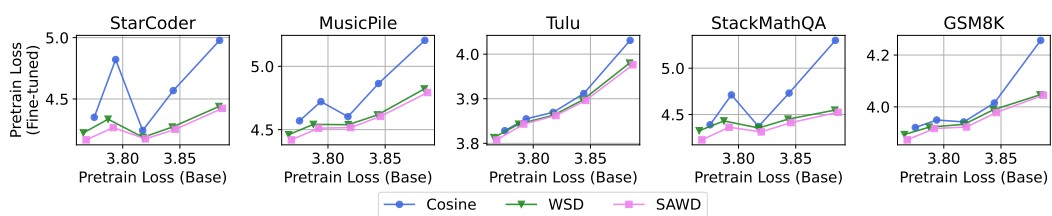

Figure 113: **WSD vs Cosine Catastrophic overtraining tradeoff for OLMo-20M across datasets**

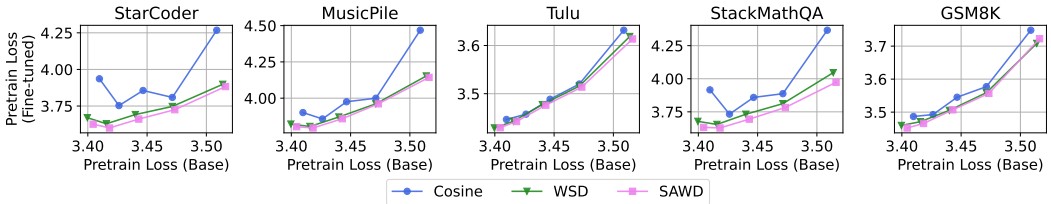

Figure 114: **WSD vs Cosine Catastrophic overtraining tradeoff for OLMo-60M across datasets**

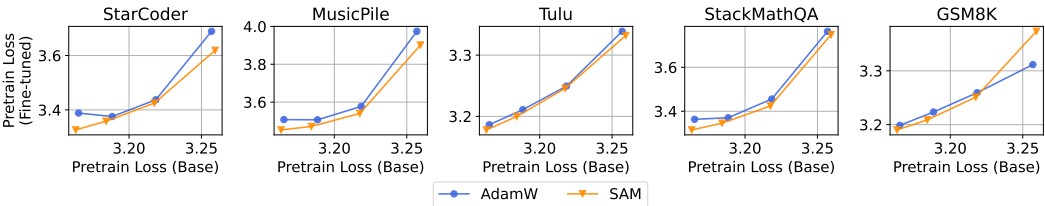

Figure 115: **AdamW vs SAM Catastrophic overtraining tradeoff for OLMo-150M across datasets**

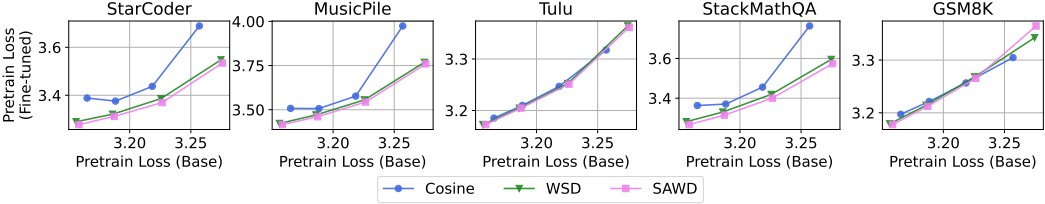

Figure 116: **WSD vs Cosine Catastrophic overtraining tradeoff for OLMo-150M across datasets**

## D.5 GAUSSIAN PARAMETER PERTURBATIONS

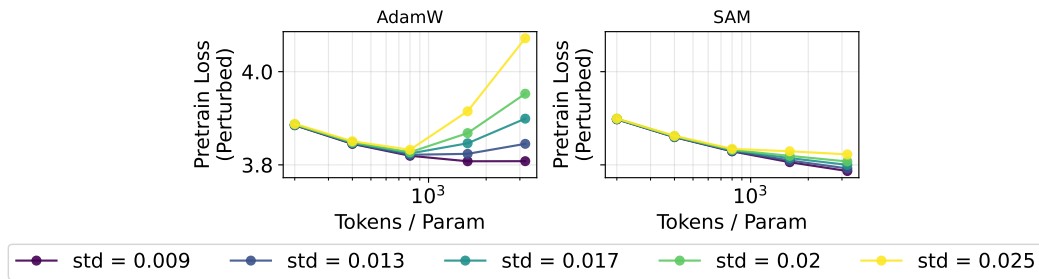

Figure 117: **AdamW vs SAM Gaussian Perturbation OLMo-20M**

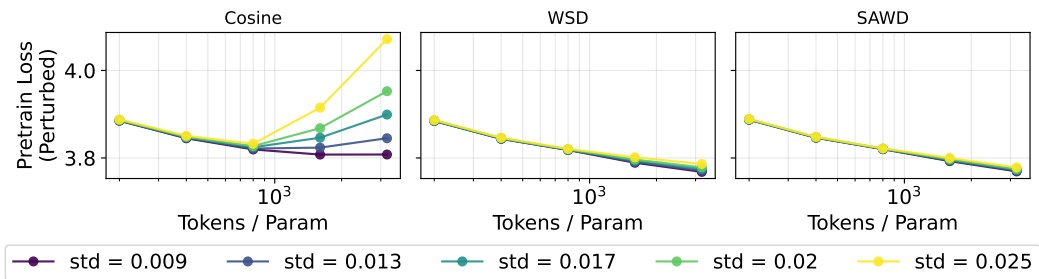

Figure 118: **WSD vs Cosine Gaussian Perturbation OLMo-20M**

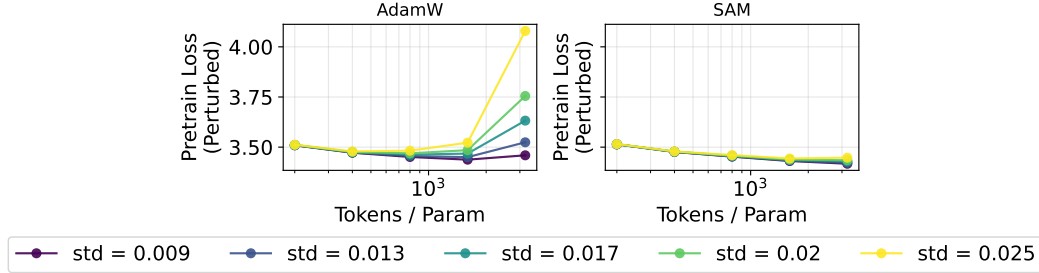

Figure 119: **AdamW vs SAM Gaussian Perturbation OLMo-60M**

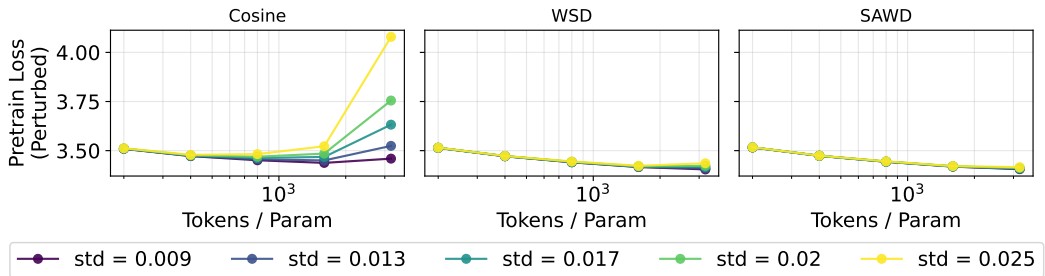

Figure 120: **WSD vs Cosine Gaussian Perturbation OLMo-60M**

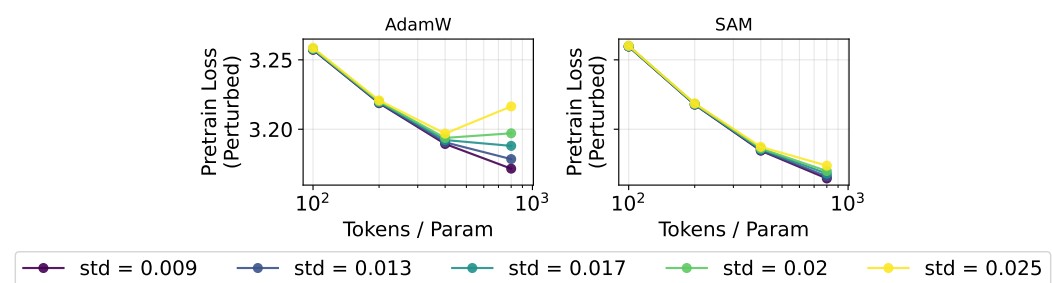

Figure 121: **AdamW vs SAM Gaussian Perturbation OLMo-150M**

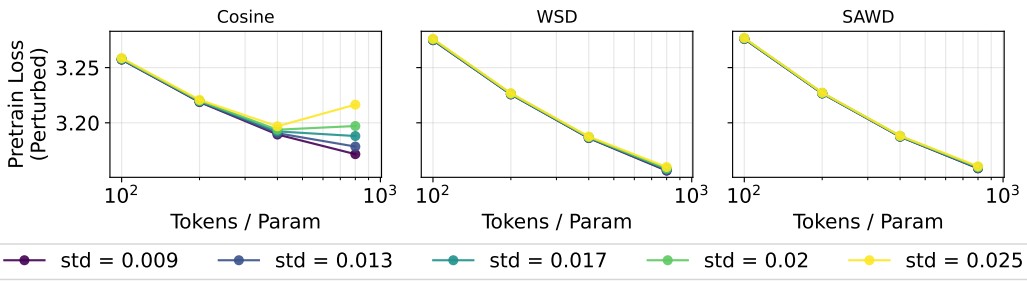

Figure 122: **WSD vs Cosine Gaussian Perturbation OLMo-150M**

## D.6 POST TRAINING QUANTIZATION

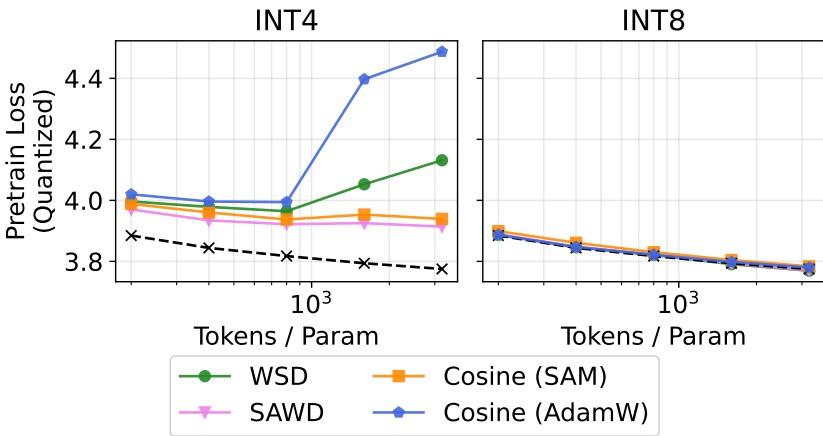

Figure 123: **Different pretraining configs PTQ performance for OLMo-20M**

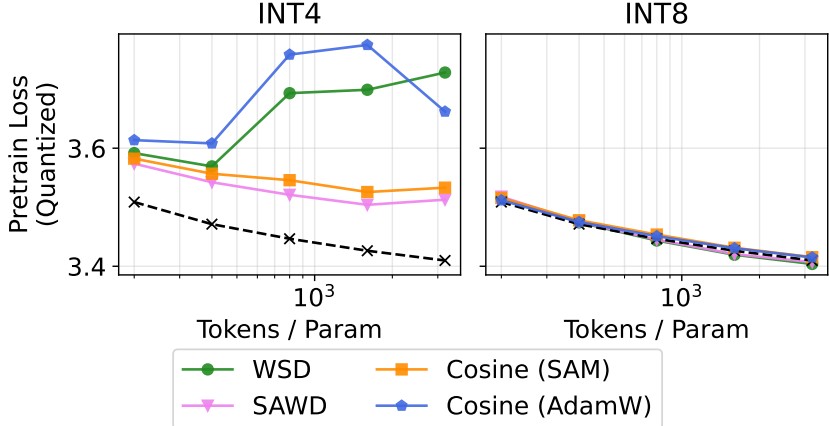

Figure 124: **Different pretraining configs PTQ performance for OLMo-60M**

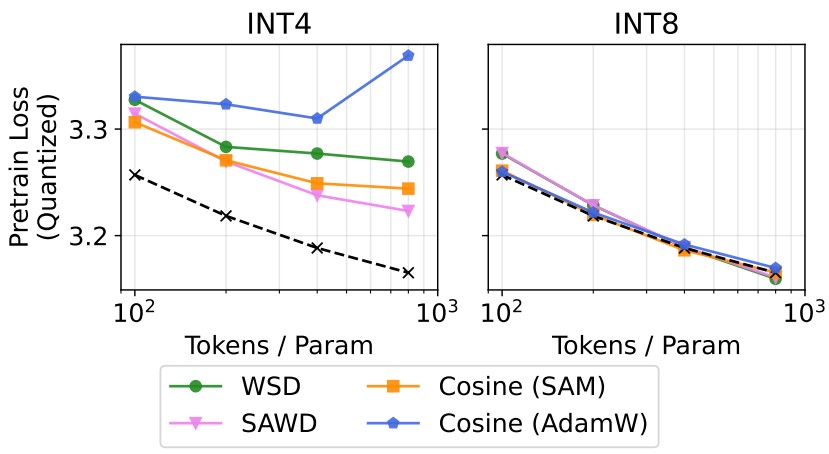

Figure 125: **Different pretraining configs PTQ performance for OLMo-150M**

