# OpenReview forum: "Sharpness-Aware Pretraining Mitigates Catastrophic Forgetting"
_ICLR.cc/2026/Workshop/Sci4DL — Sci4DL 2026_

### Official Review · Reviewer_PeuV · 2026-02-11

**Fit:** 2
**Significance:** 1
**Confidence:** 2

**Summary:**

This paper shows that language models pretrained with sharpness-reducing strategies can result in better learning-forgetting trade-offs when finetuning on downstream tasks, as well as better robustness to weight quantization and perturbation.

Two such strategies, one explicit (SAM) and one implicit (LR scheduling) are considered for 20M-150M OLMo models pretrained on 4B-192B tokens of DCLM-Baseline, and finetuned on 5 different downstream tasks including StarCode and GSM8K.

Empirically, the pareto frontier of pretraining/finetuning losses improves with these strategies, indicating better learning-forgetting trade-offs when finetuning on these downstream tasks.
Similarly, performance degradation after weight quantization or perturbation is reduced with these strategies.
Combining both strategies, the paper proposes SAWD which applies SAM during the decay phase of WSD learning rate scheduling.
Results for SAWS typically improve over results for SAM or WSD individually.

As an explanation and underlying theory for why sharpness-reducing strategies would mitigate forgetting, the paper argues that forgetting (i.e. positive change in pretraining loss) is dominated by the second order term of the pretraining loss taylor expansion (i.e. sharpness along the weight update from finetuning).

**Strengths:**

**Significance**
The empirical results are interesting and plausibly significant, showing that different methods for LLM pretraining can improve learning and mitigate forgetting when finetuning on different tasks. Related is the result that better trade-offs in final performance are possible despite worse pretraining performance.

To what extent these results are significant and generalize beyond the particular pretraining and finetuning setup of this paper is arguably up for question; but I do believe the current results and ablations are comprehensive enough, and the likelihood of significance plausible enough, for a workshop submission.

I also find these empirical results scientifically interesting in and of themselves, even if the argued relationship between sharpness and forgetting, used to motivate the experiments and explain the results, is not particularly sound or substantiated, and likely incorrect in my opinion.

**Quality**
The empirical experiments and analyses are of generally high quality, with comprehensive ablations and reporting of results (maybe 36 pages of plots in the appendix is a bit much and could be condensed).
The proposed method of combining SAM with WSD learning rate scheduling is also simple and empirically effective.
Focusing on learning/forgetting trade-offs was, I think, a good idea that reveals an important blindspot in a lot of LLM research, where pretraining performance is assumed to correlate with downstream performance, and methods that do not improve pretraining performance are maybe not explored to their full potential.

**Soundness**
I am reasonably convinced by the soundness of the empirical results, particularly with the extensive experiments and hyperparameter tuning reported in the appendix.

In terms of the motivating hypothesis and explanation relating sharpness to forgetting, I believe it has the potential to be made sound, even if in the current state of the paper, it is too problematic for me to recommend the paper for acceptance.

**Suggestions:**

**More careful and rigorous treatment of central hypothesis**
While the results are intrinsically significant and interesting (and would be great for a workshop!), I believe the general premise and framing of the paper is fundamentally flawed and likely incorrect in a significant way that needs to be more carefully addressed before the paper should be accepted.

Specifically, the idea that sharpness is an underlying cause for forgetting during finetuning is central to the paper, but casually hand-waved in Section 2 with an inadequate approximation and set of assumptions that are questionable, unsubstantiated, and (I believe) false with high likelihood.

I want to be clear here, for me, a falsified hypothesis would not be grounds for rejection in and of itself.
However, framing an entire paper and its contributions in terms of a dubious hypothesis that is hand-waved with apparently no meaningful:
1. formalization of the hypothesis;
2. theoretical justification of the hypothesis;
3. critical examination or empirical validation of the assumptions in the hypothesis; or
4. empirical validation of the hypothesis itself.

...is particularly problematic, and unfortunately unacceptable to me, as it fundamentally goes against the idea of scientific understanding central to this workshop.
Therefore, even despite the interesting and potentially significant nature of the empirical results,  I can't recommending accepting the paper because of this perfunctory treatment of a hypothesis and explanation that is central to the paper on one hand, and very likely incorrect on the other.

**Potential issues with hypothesis/explanation that can be addressed**
First, it's very unlikely that change in weights from finetuning $\Delta \theta$  is small enough to approximate change in loss with a second-order taylor expansion (L072). This can (and should) be measured and validated empirically. Even if it turns out that $\Delta \theta$ is not small enough for this approximation to be valid, you could try and look at change in weights after a single finetuning step instead, where this approximation is much more likely to be valid. Then you could try and connect the likelihood of forgetting after one finetuning step to forgetting over all finetuning steps.

Second, the claim that "after long pre-training, the gradient is small and forgetting is dominated by curvature" (L075) is also questionable and unsubstantiated. I would even argue that its almost certainly false in the context of multiple optimizer steps, as in finetuning, where large curvature entail large changes in gradients after one step, and therefore large gradients in the next step.
You could verify this empirically, but I think it would be much more productive to consider (mis)alignment between pretraining vs. finetuning gradients and hessians:
- Forgetting in your second order approximation fundamentally only happens if FT gradients/updates are misaligned with PT gradients or hessians
- Whether FT-PT gradient misalignment (first-order term) or directional sharpness along FT gradients (second-order term) dominates forgetting can be measured empirically.
- Whether and how SAM or other methods reduce forgetting via these terms can also be measured empirically.

It is very possible that observed improvements from sharpness minimizing methods are not actually attributable to sharpness minimization, but rather some effect on gradient and/or hessian (mis)alignment.

This brings me to my third point. There is existing work showing that generalization in methods like SAM is not necessarily attributable to sharpness minimization (e.g. https://arxiv.org/abs/2206.06232, https://arxiv.org/abs/2307.11007, https://arxiv.org/abs/2405.20439).
These works further reinforce my belief that, independent of previously mentioned issues, your explanation of forgetting in terms of sharpness is likely incorrect, or at least incomplete.

To conclude, the main issue and suggestion for improvement I have is how carefully and rigorously you consider your central hypothesis/explanation. For this venue in particular, I believe that more carefully and rigorously formalizing+justifying+validating your hypothesis and underlying assumptions would have made this paper an excellent fit and easy accept, regardless of the correctness of the hypothesis.
Even just adequately formalizing+justifying such a hypothesis while leaving validation as future work would have likely been acceptable.
I hope this helps clarify not only why I ultimately decided to reject the paper despite its strengths, but also what can be done concretely to improve the main issues I saw with the paper.

---

### Official Review · Reviewer_Q5Eu · 2026-02-20

**Fit:** 3
**Significance:** 3
**Confidence:** 2

**Summary:**

The paper shows that reducing sharpness explicitly (SAM) or implicitly (large learning rates and Warmup–Stable–Decay schedules) yields better tradeoff between pre-training loss and fine-tuning loss. Combining these approaches (SAWD) preserves most benefits of SAM with minimal overhead, consistently improving downstream performance.

**Strengths:**

- The paper addresses the important problem of catastrophic forgetting after fine-tuning, and experiment results demonstrate that the proposed approach alleviates this issue effectively.

- The message that sharpness-aware choices improve adaptation is intuitive and likely to inspire more work along this direction.

- The experiments are extensive and systematic.

- The writing is clear and concise, which makes the paper pleasant to read.

**Suggestions:**

- The sharpness–forgetting link is primarily empirical, and some theoretical backing could help make it more convincing.

---

### Official Review · Reviewer_N3Sp · 2026-02-26

**Fit:** 2
**Significance:** 3
**Confidence:** 3

**Summary:**

The paper empirically investigates the effectiveness of sharpness-reducing strategies applied during pre-training to improve model stability in subsequent fine-tuning or quantization. The authors demonstrate that both Sharpness-Aware Minimization (SAM) and the Warmup-Stable Decay (WSD) learning rate schedule lead to more stable solutions, with increased robustness to perturbations and improved learning-forgetting frontiers for fine-tuning, particularly in overtraining regimes. Based on these results, the authors propose the combined Sharpness-Aware Warmup-stable Decay (SAWD) training schedule, which uses the WSD learning rate schedule and applies SAM at the last decay stage to achieve the sharpness-reduction benefits of SAM without incurring most of its computational overhead.

**Strengths:**

- I find the focus on pre-training aimed at improving fine-tuning results, rather than solely reducing pre-training loss, both important and under-explored. Using sharpness-reducing strategies for this purpose is well motivated, supported by prior findings that overtraining increases the sensitivity of pre-trained models to subsequent modifications.
- The paper presents a thorough empirical study, covering a variety of model sizes, training regimes, fine-tuning tasks, and perturbations. The appendix provides detailed results for all conducted experiments.
- The paper is clearly written and easy to follow.
- I particularly appreciate that experiments comparing Adam and SAM were conducted under different matching strategies, especially the one matching the loss after pre-training.

**Suggestions:**

Technical comments on the current results/text:
* Even though the paper provides all results in the appendix, the main text would benefit from a more thorough discussion of the findings. Currently, the results presented in the main text focus on the best-performing cases for large models, large training token budgets, and two tasks (Figure 1), whereas results for smaller models, smaller budgets, and other fine-tuning tasks in the appendix generally look less positive. The effects of model size and training budget are well explained in the appendix, and highlighting these in the main text would make the discussion more complete. Similarly, the variation in effectiveness across different fine-tuning tasks could be clarified with additional analysis.
* It would be helpful to clarify which gradients are used as the base in SAM. As far as I understand, SAM can be applied on top of any first-order method (https://arxiv.org/abs/2106.01548), hence I would expect the paper to use SAM on top of AdamW, as SAM on top of SGD is generally less effective. This information does not appear to be specified in the text.
* The section on the effect of learning rate values and schedules on sharpness and forgetting (Section 2.3) would benefit from a clearer discussion of which effects are expected.
* It would be useful to include an analysis of the computational overhead of SAM and SWAD. Currently, the results for SWAD do not appear particularly strong in terms of loss, but emphasizing that it is much cheaper than SAM would make the argument more convincing.
* The full results in the appendix would benefit from improved organization, for example by grouping similar plots together within the same figure. The current presentation, with many similar plots, makes analysis more difficult.

Broader comments:
* While the empirical results in the paper are promising, a deeper understanding of these findings would strengthen the work. Although the motivation for using sharpness-reducing strategies is reasonable, additional analysis, particularly direct measurements of the sharpness of the obtained solutions, would help clarify to what extent the observed effects are truly connected to sharpness reduction. In addition, the theoretical analysis in Section 2.2 is somewhat simplified. In practical LLM pre-training, the first-order term may still play a significant role, both because pre-training loss is typically not fully optimized (so gradients are not close to zero) and because fine-tuning updates can be large enough to make the Taylor approximation less accurate. Clarifying these assumptions and their practical relevance would improve the theoretical discussion.
* From a practical perspective, the paper would benefit from larger-scale experiments, particularly since SAM and WSD show more pronounced positive effects with larger models and training token budgets.
* A more thorough related work section would strengthen the paper. In particular, the following works appear highly relevant in relation to SWAD: http://arxiv.org/abs/2206.06232, https://arxiv.org/abs/2410.10373.

---

### Meta-Review · Area_Chair_8SXz · 2026-03-02

**Recommendation:** Accept

**Metareview:**

The paper shows that optimizers that reduce the sharpness of the loss landscape during LLM pretraining yield better downstream performance after fine-tuning, reducing catastrophic forgetting. More generally, the models become more robust to weight perturbations (like fine-tuning and quantization). The work provides an extensive empirical study of this phenomenon across various models and datasets.
Although a thorough theoretical analysis of this technique is not given, the strong and promising empirical analysis makes this paper a good fit for the workshop.

---

### Decision · Program_Chairs · 2026-03-02

Accept